Corrected: Publisher correction

# Deacetylation of serine hydroxymethyl-transferase 2 by SIRT3 promotes colorectal carcinogenesis

Zhen Wei[1], Jinglue Song[2,3], Guanghui Wang[2,3], Ximao Cui[2,3], Jun Zheng[1], Yunlan Tang[1], Xinyuan Chen[1], Jixi Li [1], Long Cui[2,3], Chen-Ying Liu[2,3] & Wei Yu [1,2]

The conversion of serine and glycine that is accomplished by serine hydroxymethyltransferase 2 (SHMT2) in mitochondria is significantly upregulated in various cancers to support cancer cell proliferation. In this study, we observed that SHMT2 is acetylated at K95 in colorectal cancer (CRC) cells. SIRT3, the major deacetylase in mitochondria, is responsible for SHMT2 deacetylation. SHMT2-K95-Ac disrupts its functional tetramer structure and inhibits its enzymatic activity. SHMT2-K95-Ac also promotes its degradation via the K63-ubiquitin–lysosome pathway in a glucose-dependent manner. TRIM21 acts as an E3 ubiquitin ligase for SHMT2. SHMT2-K95-Ac decreases CRC cell proliferation and tumor growth in vivo through attenuation of serine consumption and reduction in NADPH levels. Finally, SHMT2-K95-Ac is significantly decreased in human CRC samples and is inversely associated with increased SIRT3 expression, which is correlated with poorer postoperative overall survival. Our study reveals the unknown mechanism of SHMT2 regulation by acetylation which is involved in colorectal carcinogenesis.

[1] State Key Laboratory of Genetic Engineering and Collaborative Innovation Center for Genetics and Development, School of Life Sciences and Zhongshan Hospital, Fudan University, Shanghai 200438, China. [2] Department of Colorectal and Anal Surgery, Xinhua Hospital, Shanghai Jiao Tong University School of Medicine, Shanghai 200092, China. [3] Shanghai Colorectal Cancer Research Center, Shanghai 200092, China. These authors contributed equally: Zhen Wei, Jinglue Song. Correspondence and requests for materials should be addressed to L.C. (email: cuilong@xinhuamed.com.cn) or to C.-Y.L. (email: liuchenying@xinhuamed.com.cn) or to W.Y. (email: yuw@fudan.edu.cn)

One-carbon metabolism not only provides cellular components including nucleotides, lipids and proteins for cell growth but also generates glutathione and S-adenosylmethionine, which are needed to maintain the cellular redox status and epigenetic status of cells[1]. The role of one-carbon metabolism in tumorigenesis has been extensively studied[2–4], and the antagonism of one-carbon metabolic enzymes has been used in chemotherapy for over 60 years[5]. Serine and glycine, two nonessential amino acids, are major inputs for one-carbon metabolism and are used for nucleotide synthesis. Recently, disorders of serine and glycine metabolism during carcinogenesis have gained attention[6]. A key serine/glycine conversion enzyme whose expression is consistently altered during tumorigenesis is serine hydroxymethyltransferase (SHMT). SHMT is the enzyme that catalyzes the reversible conversion of serine to glycine via the transfer of the β-carbon of serine to tetrahydrofolate (THF), and this conversion resulting in the formation of 5,10-methylene-THF and glycine; these in turn are involved in the folate cycle. Two SHMT genes, SHMT1 and SHMT2, have been identified in the human genome. SHMT1 encodes the cytoplasmic isozyme involved in the de novo synthesis of thymidylate[7], while SHMT2, which encodes the mitochondrial isozyme, participates in the synthesis of mitochondrial thymidine monophosphate (dTMP)[8]. Strikingly, SHMT2 but not SHMT1 expression is significantly upregulated in a variety of cancers, including colorectal, brain, central nervous system (CNS), kidney, and bladder cancers[9,10]. Two clinical studies have shown that high expression of SHMT2 is associated with tumor aggressiveness and prognosis[11,12]. In breast cancer, HIF1α and MYC cooperate to drive SHMT2 upregulation, which leads to an increased concentration of nicotinamide adenine dinucleotide phosphate (NADPH) and enhanced redox balance; this in turn facilitates cancer cell growth under hypoxic conditions[10]. However, whether post-translational modification affects the level of SHMT2 protein in tumorigenesis and how the upregulation of SHMT2 is involved in colorectal carcinogenesis are unknown.

Two protein lysine modifications, acetylation and ubiquitination, are coordinately regulated to control critical cellular functions. Several metabolic enzymes are regulated by acetylation through ubiquitin-dependent proteasome degradation or lysosomal-dependent degradation[13]. In this study, we report that the activity and protein stability of the mitochondrial metabolic enzyme SHMT2 are regulated by lysine acetylation. Specifically, acetylation of lysine K95 inhibits SHMT2 activity and promotes K63-Ub-lysosome-dependent degradation of SHMT2 via macroautophagy. We investigated the functional significance of SHMT2 expression and acetylation levels in colorectal tumorigenesis. Our study reveals the previously unknown mechanism of SHMT2 regulation by acetylation in the one-carbon metabolic pathway that is involved in colorectal carcinogenesis.

## Results

**SHMT2 is acetylated at K95.** Recent mass spectrometry-based proteomic analyses have identified a large number of potentially acetylated proteins, including SHMT2[14]. To confirm the acetylation of SHMT2 in vivo, Flag-tagged SHMT2 was ectopically expressed in HeLa cells and immunoprecipitated. Western blot with an anti-pan-acetyl-lysine antibody confirmed that SHMT2 was indeed acetylated and that its acetylation was enhanced approximately two-fold after treatment with nicotinamide (NAM, an inhibitor of the sirtuin (SIRT) family of deacetylases)[15] (Fig. 1a). Similar experiments in human osteosarcoma U2OS cells also showed that NAM treatment enhanced SHMT2 acetylation (Fig. 1a). In one of our previously published papers, we reported that acetylation at K464 of SHMT2 was increased by 4.7-fold in

Sirt3$^{-/-}$ MEFs compared with Sirt3$^{+/+}$ MEFs[14]. In addition, K280 in the catalytic domain of SHMT2 was identified by an acetylation proteomics study[16]. To test whether these two sites are primary acetylation sites, we generated Arg (to mimic deacetyl-modification) and Gln (to mimic acetyl modification)[17–19] substitution mutants of both sites (K280R, K280Q, K464R, K464Q). However, none of the mutants influenced the overall acetylation level of SHMT2 (Supplementary Fig. 1a), which indicates that neither K464 nor K280 is the major acetylation site of SHMT2 in our study. Moreover, the SHMT2 K464R/Q mutant exhibited an activity similar to that of the wild-type (WT) protein, while the K280R/Q mutant exhibited no activity due to disruption of the active site, which also suggests that our method of detection of SHMT2 activity is feasible (Supplementary Fig. 1b). To investigate the functional acetylated regulatory sites of SHMT2, mass spectrometry analysis was performed using Flag-tagged SHMT2-expressing stable cells. Lys95 of SHMT2 was found to be acetylated (Fig. 1b). Lys95 in SHMT2 is highly conserved in different species from E. coli to Homo sapiens (Fig. 1c), indicating the potential critical role for K95 in the function of SHMT2. Western blot with an anti-pan-acetyl-lysine antibody showed that the K95R/Q mutant exhibited significantly reduced overall acetylation of SHMT2 in HeLa cells (Fig. 1d), showing that K95 is the major acetylation site in SHMT2. To confirm K95 acetylation in vivo, we generated an antibody that specifically recognizes acetylated K95 in SHMT2. Using this site-specific antibody, a strong signal for ectopically expressed WT SHMT2 was detected by western blot, but no signal for the K95Q mutant was observed in SHMT2 knockout cells (Fig. 1e). The specificity of this antibody was also verified as it recognized the K95-acetylated peptide but not the unacetylated control peptide (Supplementary Fig. 1c). These results demonstrated the specificity of our SHMT2-K95-Ac antibody in its recognition of SHMT2-K95-Ac. Furthermore, western blot of whole-cell extract from different types of cancer cells detected a band with the expected molecular weight of SHMT2 and whose intensity was substantially increased upon treatment of cells with NAM (Fig. 1f). These results demonstrate that SHMT2 is primarily acetylated at K95 and that K95 acetylation occurs in different cancer cells.

**SIRT3 is the major deacetylase for SHMT2.** Since NAM treatment increased SHMT2 acetylation, we sought to identify which nicotinamide adenine dinucleotide (NAD$^+$)-dependent sirtuin(s) are involved in SHMT2 deacetylation. Given that SHMT2 is localized in the mitochondria, we examined whether the major mitochondrial deacetylase SIRT3[20] could deacetylate SHMT2 and affect its function. We found that SHMT2 directly interacted with SIRT3 (Fig. 1g and Supplementary Fig. 1d). Then, we performed an in vitro SHMT2 deacetylation assay to determine whether SIRT3 directly deacetylates SHMT2. A decrease in the level of acetylated SHMT2 was observed following the addition of recombinant human (rh) SIRT3 in the presence of NAD$^+$, which was reversed by the addition of nicotinamide (NAM) (Fig. 1h). Thus, SIRT3 plays a role in SHMT2 deacetylation. We next investigated the role of SIRT3 in SHMT2 deacetylation at the cellular level. In cells that overexpressed SIRT3, the level of acetylated SHMT2 was significantly lower than that in cells containing the empty vector (Fig. 1i). We also performed an in vitro SHMT2-K95-Ac deacetylation assay. As expected, deacetylation of SHMT2-K95-Ac by SIRT3 strictly depended on the presence of NAD$^+$ and was completely blocked by NAM treatment. In addition, the catalytically inactive SIRT3-H248Y mutant[21] did not deacetylate SHMT2 despite the presence of NAD$^+$ (Fig. 1j). Furthermore, SHMT2-K95-Ac was increased in

SIRT3 knockout colon cancer cells (Fig. 1k), and this outcome was also confirmed in cells in which SIRT3 was stably knocked down (Supplementary Fig. 1e). This knockout resulted in decreased enzymatic activity and indicates that acetylation of SHMT2 can negatively regulate its enzymatic activity. Moreover,

we tested the SHMT2-K95-Ac levels in *Sirt3*−/− MEFs and found that SHMT2-K95-Ac was increased in *Sirt3*−/− MEFs compared with wild-type MEFs. Interestingly, we also found the level of endogenous SHMT2 was significantly decreased in *Sirt3*−/− MEFs (Fig. 1l) and SIRT3 knockdown cells (Supplementary

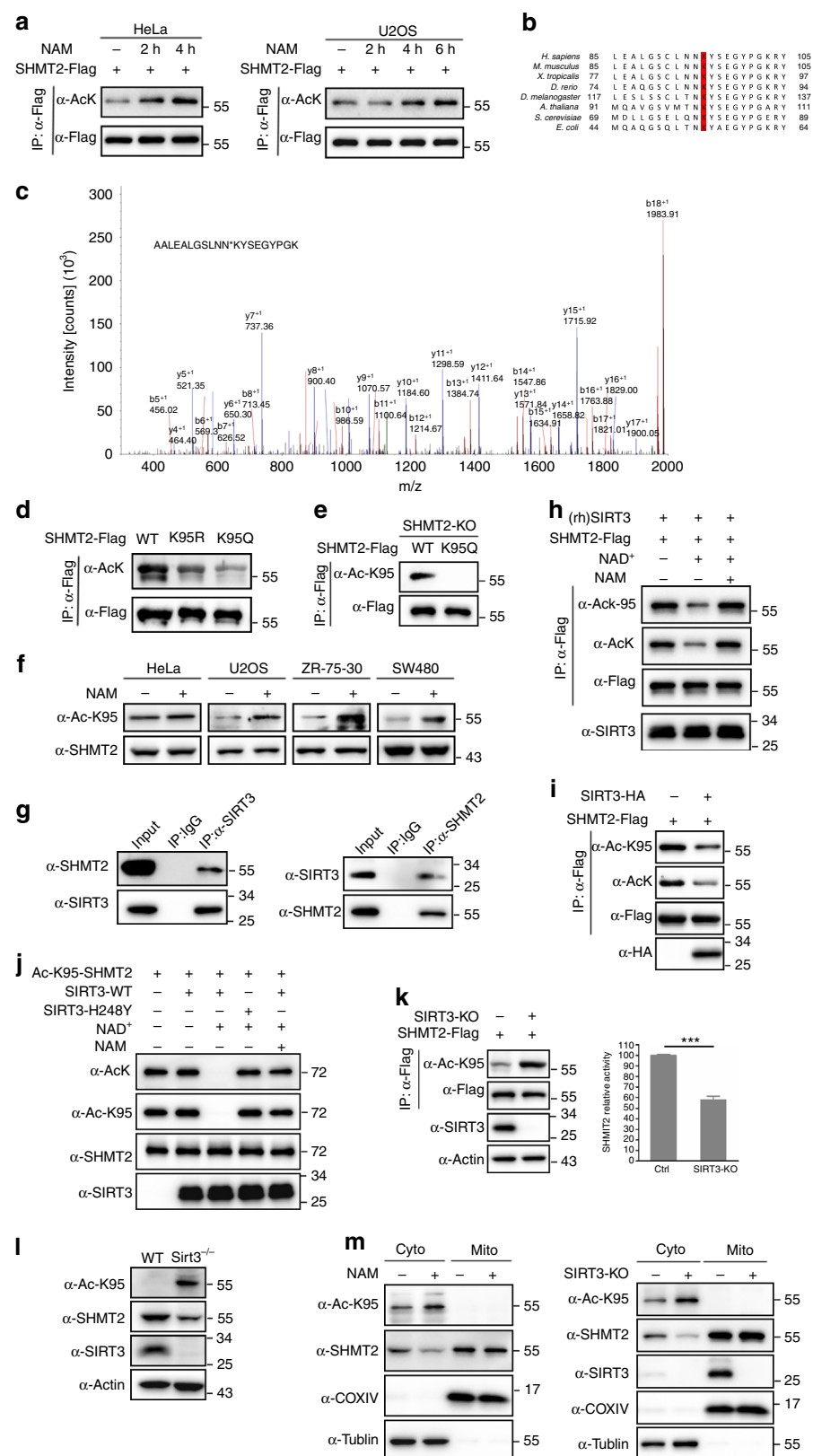

Fig. 1e), but mRNA level of SHMT2 was mildly increased in SIRT3 knockdown cells (Supplementary Fig. 1e), indicating that acetylated K95 may affect the stability of SHMT2 protein. Finally, we performed a subcellular fractionation assay, which involved the treatment of HCT116 cells with NAM or the knockout of SIRT3, to determine the localization of SHMT2-K95-Ac. An SHMT2-K95-Ac was strongly elevated in the cytoplasm of HCT116 cells after treatment with NAM and after SIRT3 knockout (Fig. 1m), indicating that acetylated K95 may be involved in SHMT2 degradation in the cytoplasm. Collectively, these data suggest that SIRT3 is the major deacetylase of SHMT2 and is responsible for the deacetylation of K95.

**K95 acetylation impairs the formation of tetrameric SHMT2 and inhibits its enzymatic activity.** To confirm whether K95 acetylation negatively regulates the enzymatic activity of SHMT2, we overexpressed the K95R and K95Q mutants of SHMT2 in HeLa cells and compared the proteins with WT SHMT2. We found that the mutation of Lys95 to arginine reduced SHMT2 activity by 63%. Notably, the mutation of Lys95 to glutamine dramatically decreased SHMT2 activity by as much as 95% (Fig. 2a). Furthermore, NAM treatment decreased SHMT2 activity by 83% in HeLa cells (Fig. 2b). To investigate the mechanism by which Lys95-acetylation reduces SHMT2 activity, we expressed and purified recombinant human WT SHMT2 as well as the K95R and K95Q mutants from *E. coli*. Steady-state kinetic analysis of SHMT2 and the variants (Fig. 2c) indicated that substitution of Lys95 with arginine or glutamine decreased the $V_{max}$ so dramatically that the $K_m$ values for DL-β-phenylserine, which was used as the substrate for SHMT2 to produce benzaldehyde and is detectable by its strong ultraviolet (UV) absorbance at 279 nm[22], could not be effectively detected. However, as a control, the K464Q variant exhibited similar $V_{max}$ and $K_m$ values as WT SHMT2. To definitively demonstrate the effect of K95 acetylation on SHMT2 activity, we employed genetically encoded Nε-acetyl-lysine to prepare recombinant proteins in *E. coli*[23,24]. This expression system produced SHMT2 proteins with 100% acetylation at K95 due to the suppression of the K95-TAG stop codon by the Nε-acetyl-lysine-conjugated amber suppressor tRNA. Like the acetyl-mimetic modification in the mutation of Lys to Gln, the SHMT2-K95-Ac exhibited no enzymatic activity (Fig. 2d). Collectively, these results demonstrate that acetylation at lysine 95 inhibits SHMT2 activity. However, the way in which K95 acetylation inactivates this enzyme and whether its conformation is changed remains unknown.

We attempted to resolve the crystal structures of SHMT2 and its mutants to reveal the potential molecular mechanism. Luckily, Giorgio Giardina and his colleagues resolved the crystal structure of WT human SHMT2 in 2015[25]. Their research served as a good reference for our work. Indeed, we determined the structure of the K95R mutant at a resolution of 3.0 Å (Supplementary Table 1) but failed to determine the structure of the K95Q mutant due to its ease of precipitation. Compared with K95 (PDB ID 4PVF), R95 (PDB ID 5 × 3V) has more hydrogen bond interactions with its binding partner (Fig. 2e), which abolishes the formation of functional tetramers. Additionally, the molecular dynamic simulations of the K95Q mutant show that Q95 has fewer hydrogen bond interactions with its binding partner (Fig. 2e), and thus, the K95Q mutant may exist as a monomer; this outcome was confirmed by gel filtration assay (Fig. 2f and Supplementary Fig. 2). In mammals, SHMT2 is active only as a homotetramer, whereas in *E. coli* it is active as a dimer[25,26]. Altogether, this structural analysis indicates that both the R and Q mutants can not only lose their ability to form a more effective tetramer, but they also form more or less hydrogen bonds, causing the loss of appropriate space for substrate insertion. In turn, this results to exhibit low enzymatic activity. Using the SHMT2 mutants tagged by either Flag or Myc, we found that the K95R and K95Q mutants exhibited an attenuated ability to interact with each other (Fig. 2g), which is similar to polymer conformation destruction. Finally, to confirm that acetylated K95 affects the multimerization status of cellular SHMT2, we performed glutaraldehyde crosslink experiments in SHMT2 knockout or SIRT3 knockout HCT116 cells. The tetramer status of the K95Q mutant was disrupted in HCT116 cells compared with wild-type SHMT2 (Fig. 2h). The tetramer status of acetylated SHMT2 is also significantly decreased in SIRT3 knockout HCT116 cells (Fig. 2h). These data further demonstrate that acetylated SHMT2-K95 destroys SHMT2 function by disrupting the tetramer status. Taken together, these data indicate that the acetylation of K95 on the dimer interface disrupts its SHMT2 functional tetramer conformation and inactivates the enzyme.

**K95 acetylation promotes SHMT2 degradation through macroautophagy.** In addition to its effect on SHMT2 enzyme activity, NAM treatment also led to a time-dependent reduction in SHMT2 protein levels but not SHMT2 mRNA levels (Fig. 3a and Supplementary Fig. 3a, b), and this reduction was accompanied by a time-dependent increase in SHMT2-K95-Ac. This is consistent with the finding that endogenous SHMT2 protein was decreased in *Sirt3*$^{-/-}$ MEFs and in cells in which SIRT3 was

**Fig. 1** SHMT2 is acetylated at K95 and SIRT3 is the major deacetylase for SHMT2. **a** Western blot detection of acetylation levels of ectopically expressed SHMT2 in HeLa and U2OS cells after treated with 7.5 mM NAM for the duration indicated. AcK, pan-acetyl-lysine antibody. **b** Acetylated SHMT2 K95 was identified by a tandem mass spectrum. The identified peptide is shown. **c** K95 in SHMT2 is conserved. The sequences around SHMT2 K95 from different species were aligned. **d** K95 is the primary acetylation site of SHMT2. The indicated plasmids were transfected into HeLa cells, the acetylation levels were analysis by western blot. **e** Characterization of acetyl-SHMT2 (K95) antibody. Acetylation level of SHMT2-Flag or SHMT2-K95Q-Flag ectopically expressed in SHMT2-knockout HCT116 cells was measured by the site-specific K95 acetylation antibody (Ac-K95). **f** K95 acetylation of SHMT2 is broadly verified in different human cancer cells. Endogenous SHMT2 K95-Ac levels of indicated cells after NAM treatment. **g** SIRT3 interacts with SHMT2 in vivo. Whole HCT116 cell lysates were immunoprecipitated with control IgG, anti-SIRT3 or anti-SHMT2 antibodies, and precipitated proteins were detected by anti-SHMT2 or anti-SIRT3 antibodies, respectively. **h** SIRT3 deacetylates SHMT2 in vitro. Recombinant human (rh) SIRT3 deacetylates SHMT2 proteins from HCT116 cells. **i** The acetylation levels of IPed SHMT2 were determined by western blot in stable SIRT3-overexpressing HCT116 cells. **j** Recombinant SIRT3 wild-type or a catalytically inactive mutant H248Y and the site-specific K95-acetylated SHMT2 incubated with NAD$^+$ or NAM, and acetylation level was determined by Ac-K95 and AcK. **k** SIRT3 knockout increases SHMT2 acetylation level and decreases SHMT2 activity. Corresponding SHMT2 activity was measured and normalized against protein levels. Error bars represent ± s.d. for triplicate experiments. ***$P < 0.001$ by Student's *t*-test. **l** SHMT2 K95-Ac increases and protein level decreases in *Sirt3*$^{-/-}$ MEFs. Western blot detection of endogenous SIRT3 and SHMT2 protein and K95-Ac. **m** NAM treatment or SIRT3 deficiency increases SHMT2-K95-Ac and decreases SHMT2 protein level in cytoplasm. Cytosolic and mitochondria fractions of HCT116 cells after NAM treatment for 6 h were obtained (left). Cytosolic and mitochondria fractions of SIRT3-knockout HCT116 cells were obtained (right). The levels of SHMT2 protein and SHMT2-K95-Ac were compared

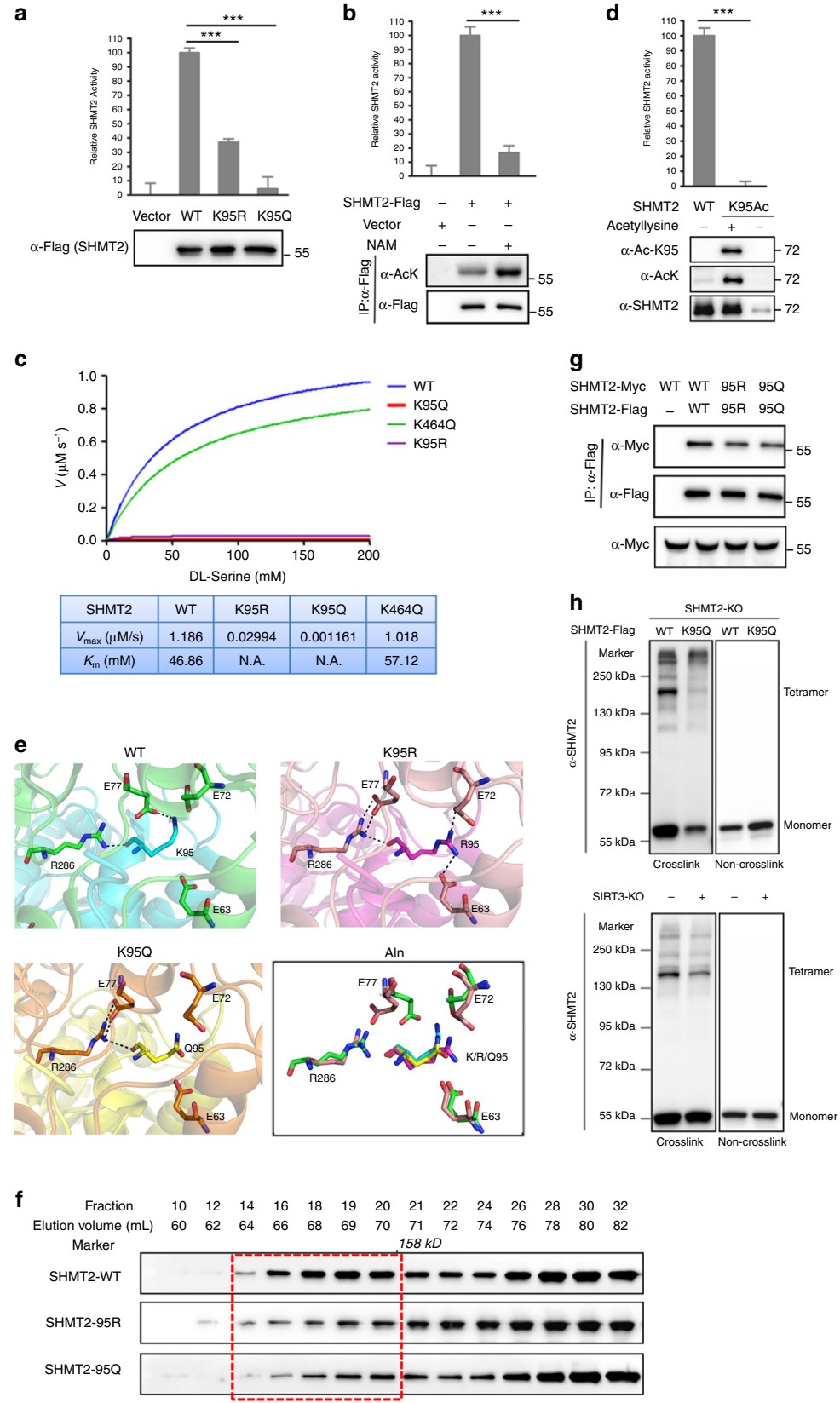

stably knocked down (Fig. 1l and Supplementary Fig. 1e). Inhibition of protein synthesis with cycloheximide (CHX) showed that SHMT2 was a relatively stable protein in SW480 cells, with a half-life longer than 8 h, but NAM treatment accelerated SHMT2 degradation under this condition (Fig. 3b). As a material for de novo serine synthesis, glucose is pivotal for cancer cell metabolism. We, therefore, tested the effect of glucose on SHMT2 K95 acetylation and protein stability. We found that K95 acetylation was increased under higher glucose concentrations in a dose-dependent manner (Fig. 3c). More interestingly, the steady-state

**Fig. 2** K95 acetylation impairs the formation of tetrameric SHMT2 and inhibits its enzymatic activity. **a** K95 mutation decreases SHMT2 enzyme activity. The enzyme activity of SHMT2 wild-type and K95R/Q mutants purified by immunoprecipitation from HeLa cells was measured and normalized against protein level. **b** Inhibition of deacetylases decreases SHMT2 enzyme activity. The enzyme activity of SHMT2 purified by immunoprecipitation from HeLa cells after NAM treatment was measured and normalized against protein level. The protein levels and acetylation levels were determined by western blot. **c** Kinetic comparison of wild-type SHMT2 and variants. Recombinant SHMT2 proteins were purified, and steady-state kinetic analyses for DL-β-phenylserine were performed. Comparison of WT (blue line) and variants K464Q (green line, negative control), K95R (violet line) and K95Q (red line) shows that the K95 is important in catalysis activity. N.A. represents not detected effectively. **d** The site-specific K95-acetylated SHMT2 is a totally no activity protein.SHMT2 and K95Ac-SHMT2 were recombinant expressed and detected in total lysates by western blot or purified by nickel affinity chromatography and performed activity assay. **e** Molecular modeling of acetylation of K95 in SHMT2. **f** K95 mutation prevents SHMT2 tetramerization. Recombinant wild-type and K96R/Q mutant SHMT2 proteins were purified by nickel affinity chromatography and separated by gel filtration, followed by western analysis. Dotted box denotes K95R/Q mutant has less tetrameric formation fractions. **g** K95 mutation attenuates the interaction of SHMT2 monomers. Flag-tagged and Myc-tagged SHMT2 WT or K95R/Q mutants were expressed in HEK293T cells, and the proteins were immunoprecipitated before being subjected to western blot. **h** K95Q mutation or loss of SIRT3 disturbs SHMT2 tetramerization. Cell lysates of SHMT2-knockout HCT116 cells expressing Flag-tagged WT or K95Q mutant SHMT2 (up) and cell lysates of SIRT3-knockout or control HCT116 cells (down) were treated with or without 0.025% glutaraldehyde and analyzed by western blot with anti-SHMT2 antibody. Tetramer and monomer of SHMT2 were indicated. For **a**, **b**, and **d**, mean values ± s.d. of relative enzyme activity of triplicate experiments are presented. ***$P < 0.001$

level of SHMT2 decreased with the increase in SHMT2-K95-Ac and the decreased SIRT3 level, and this outcome is consistent with previous reports that stated that high glucose inhibits SIRT3 expression (Fig. 3d). Moreover, glucose starvation increased the interaction between SHMT2 and SIRT3 (Fig. 3e). These results indicate that low glucose increases the level of SIRT3 protein and its interaction with SHMT2, leading to the deacetylation of SHMT2 K95 and an increase SHMT2 protein stability. In our study, treatment with the proteasome inhibitor MG132 did not increase SHMT2 protein levels (Supplementary Fig. 3c). These results indicate that the acetylation-induced decrease of SHMT2 expression is mediated by a mechanism that is independent of proteasomes.

Most proteins with long half-lives are degraded by autophagic pathways within lysosomes[27,28]. We, therefore, treated cells with ammonium chloride ($NH_4Cl$) to inhibit lysosomal proteolysis that is dependent on autophagy; this system delivers substrates to lysosomes, and we found that this treatment markedly inhibited endogenous SHMT2 degradation (Fig. 3f), indicating that SHMT2 degradation depends on lysosomal autophagy. Autophagy has three main forms: macroautophagy, microautophagy, and chaperone-mediated autophagy (CMA)[29]. Macroautophagy has the capacity to engulf large structures through both selective and nonselective mechanisms[30]. In selective macroautophagy, insoluble proteins that are tagged with K63 poly-Ub chains were found to be conducive to aggresome formation[31]. Recently, SHMT2 was identified as a component of the BRCC36 isopeptidase complex (BRISC), which is capable of disassembling K63-Ub chains from IFNAR1, thus limiting receptor endocytosis and lysosomal degradation[32]. Moreover, we found that both the K95Q mutant and acetylated SHMT2-K95 tended to form an inclusion body when expressed in *E.coli*. These previous findings and our own experimental data suggest that K95 acetylation might increase the degradation of SHMT2 through K63-poly-Ub-dependent macroautophagy. The addition of the macroautophagy inhibitor 3-methyladenine, impeded the degradation of SHMT2 (Fig. 3g) but prolonged serum starvation, which is known to activate CMA[33,34], did not promote the degradation of SHMT2 (Supplementary Fig. 3d). Moreover, we specifically knocked down Atg5 or Atg7 and found that SHMT2 had accumulated (Fig. 3h and Supplementary Fig. 3e). Meanwhile, we detected another mitochondrial core folate enzyme MTHFD2 under this condition. Indeed, MTHFD2 protein level did not increase when Atg5 or Atg7 was knocked down (Supplementary Fig. 3f). Additionally, we observed that the SHMT2-K95Q mutant exhibited a strong binding affinity to p62, the macroautophagy receptor (Supplementary Fig. 3g). These data show that acetylation of SHMT2-

K95 facilitates its own degradation through lysosome-mediated-macroautophagy. Through the co-overexpression of Flag-tagged SHMT2 and HA-tagged ubiquitin in HEK293T cells, we observed that, compared with WT SHMT2, the K95Q mutant was able to bind more ubiquitin (Fig. 3i lane 2), especially K63-only ubiquitin (Fig. 3i lane 4). WT SHMT2 rarely interacted with ubiquitin in the absence of stress (Fig. 3i lane 1) and only weakly interacted with K63-only ubiquitin (Fig. 3i lane 3). In contrast to K63-only ubiquitin, K48-only ubiquitin, which targets proteins to the proteasome for degradation[35], displayed blotting bands of much lower intensity for both WT and K95Q SHMT2 (Fig. 3i lane 5 and 6). Acetylation-mimetic mutant K95Q, but not WT or K95R mutant SHMT2 (Supplementary Fig. 3h and 3i), was robustly tagged with K63-poly-Ub chains, demonstrating the importance of acetylation for this change in ubiquitin-dependent autophagic protein degradation. Treatment with the autophagy inhibitor $NH_4Cl$ increased the ubiquitin binding signal on WT and K95Q mutant SHMT2 (Fig. 3j). In addition, more acetylation modifications as a result of NAM treatment also increased SHMT2-associated K63 poly-Ub chains (Fig. 3k) to levels similar to those of the acetylation-mimetic mutant K95Q (Fig. 3i). Altogether, these data support the model in which lys95-acetylation promotes SHMT2 degradation via the K63-ubiquitin-lysosome pathway.

**TRIM21 is the E3 Ligase for SHMT2.** To identify the E3 ligase for SHMT2, we performed affinity purification coupled with mass spectrometry analysis and identified TRIM21 as a putative SHMT2-interacting protein (Fig. 4a). Interestingly, our mass spectrometry-based interactome also identified the components of BRISC complex, which is a de-K63Ub complex. This observation also indicated that SHMT2 may be degraded through the K63-ubiquitin pathway. Indeed, we verified the interaction between TRIM21 and SHMT2 in HEK293T cells (Fig. 4b). The expression of TRIM21, but not the expression of the ligase-dead (LD) mutant TRIM21 (C16A, C31A and H33W)[36], increased SHMT2 ubiquitylation (Fig. 4c), which suggests that TRIM21 is the E3 ligase for SHMT2 and that the E3 ligase activity of TRIM21 is required for SHMT2 ubiquitylation. Compared with WT SHMT2, the K95Q mutant could bind more endogenous TRIM21 (Fig. 4d). The overexpression of TRIM21 increased WT and K63-type SHMT2 ubiquitylation but not K63R-type ubiquitylation (Fig. 4e). To further confirm the positive role of TRIM21 in K63-type SHMT2 ubiquitylation, we established a stable TRIM21 knockdown HEK293T cell line. As shown in Fig. 4f, WT and K63-type SHMT2 ubiquitylation was significantly

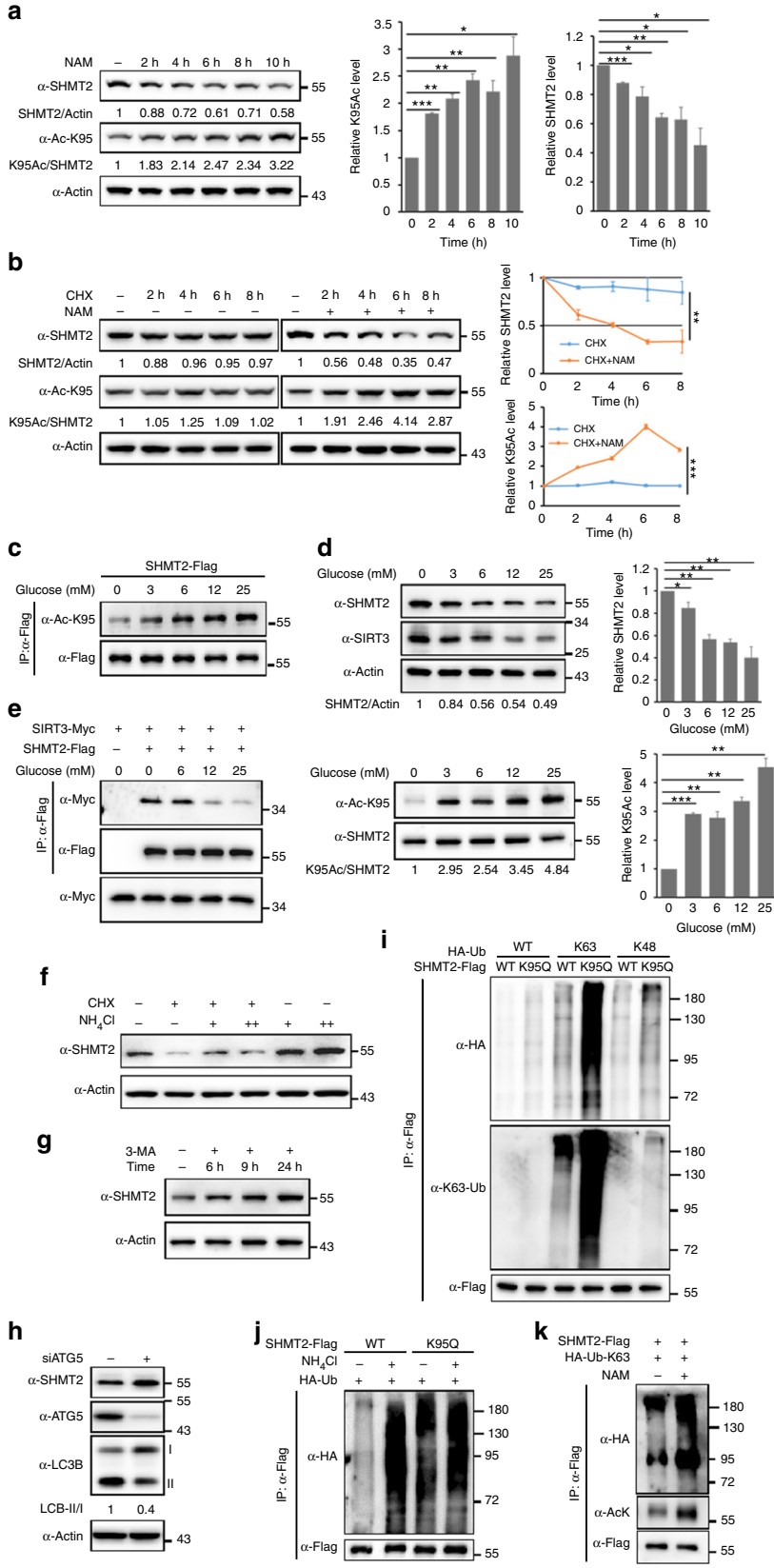

decreased with no change in the K63R-type SHMT2 in the stable TRIM21 knockdown HEK293T cells. Since higher glucose concentrations can induce SHMT2 K95 acetylation (Fig. 3c, d), we tested the effect of glucose on SHMT2 ubiquitylation. We found that the overexpression of TRIM21 increased the degradation of

SHMT2 in high glucose conditions by binding more K63-ubiquitin (Fig. 4g). Moreover, overexpression of TRIM21 can accelerate the degradation of SHMT2 in high glucose conditions (Fig. 4h). Additionally, we observed increased endogenous SHMT2 protein levels in TRIM21 knockdown HCT116 cells with

**Fig. 3** K95 acetylation promotes SHMT2 degradation through macroautophagy. **a** Inhibition of deacetylases reduces endogenous SHMT2 protein level. Western blot detection of K95-Ac and protein level of endogenous SHMT2 in SW620 cells after NAM treatment for the duration indicated. **b** SHMT2 is a stable protein and NAM treatment accelerates SHMT2 degradation. SW480 cells were treated with protein synthesis inhibitor cycloheximide (CHX) (75 µg ml$^{-1}$) with or without NAM treatment for duration indicated. Endogenous SHMT2 protein and K95-Ac level was analyzed by western blot. **c** High glucose increases SHMT2 K95-Ac. SHMT2-Flag was transfected 24 h and then cells were treated with indicated glucose concentration. **d** High glucose decreases SHMT2 protein level. Cells were cultured with different concentration of glucose. The steady-state level and K95-Ac of endogenous SHMT2 were analyzed by western blot. **e** Low glucose promotes the interaction between SIRT3 and SHMT2. The indicated plasmids were co-transfected 24 h, followed by indicated glucose concentration treatment, and the interaction between SIRT3 and SHMT2 was examined by immunoprecipitation. **f** SHMT2 is degraded by the autophagic pathways. Cells were treated with CHX for 10 h and meanwhile treated with autophagic pathways inhibitor NH$_4$Cl for 12h. NH$_4$Cl treatment, "$+$" represent 10 mM and "$++$" represent 20 mM. **g** 3-MA increases SHMT2 accumulation. Cells were treated with 3-MA (2 mM) for duration indicated. **h** ATG5-knockdown increases SHMT2 protein level. HCT116 cells were transfected with siATG5 48 h before harvest. The levels of SHMT2 and LC3B were analyzed by western blot. The decrease of LC3BII/I indicates autophagic activity suppressed. **i** K95Q mutation increases the K63-polyUb binding of SHMT2. Flag-tagged SHMT2 WT and K95Q were co-transfected with HA-Ub WT, K48-only, and K63-only mutants in HEK293T cells. **j** NH$_4$Cl increases the ubiquitin binding of SHMT2. Plasmids transfection for 18 h then cells were treated with 15 mM NH$_4$Cl for 24 h before harvest. **k** NAM treatment increases SHMT2 K63 poly-Ub chains. SHMT2-Flag and HA-Ub-K63-only were co-transfected for 36 h and then cells were treated 7.5 mM NAM for 4 h before harvest. For **a**, **b**, **d** mean values of quantitation ± s.d. are reported. *$P < 0.05$; **$P < 0.001$; ***$P < 0.001$. Representative western blot results and quantitation (herein after) of triplicated western blot are shown

no change in SHMT2 mRNA expression (Fig. 4i). Taken together, these data are consistent with the model in which TRIM21 binds acetylated SHMT2 and leads to K63-ubiquitin–lysosome degradation.

**SHMT2 acetylation inhibits cell proliferation and tumor growth**. To investigate the effect of K95 acetylation of SHMT2 on cell proliferation, shRNA-resistant WT and K95Q mutant SHMT2 were re-expressed in the SHTM2 knockdown SW480 cells at a level similar to that of endogenous SHMT2 (Fig. 5a). The knockdown SHMT2 decreased the proliferation of SW480 cells, which were substantially rescued by the re-expression of WT SHMT2 (Fig. 5a). Notably, the K95Q SHMT2 mutant was unable to restore cell proliferation in SHMT2 knockdown cells. These results demonstrate that acetylation at Lys95, which reduces the activity and stability of SHMT2, impairs the ability of SHMT2 to support CRC cell proliferation. It has been reported that high serine consumption is required for rapid proliferation of several cancer cell types[6]. To confirm that the impaired ability of the K95Q SHMT2 mutant to support SW480 cell proliferation could be due to reduced serine consumption, we measured the concentration of serine and glycine in SHMT2 knockdown cells in which either WT or K95Q mutant SHMT2 was re-expressed. We found that the serine level and the serine/glycine ratio were increased in SHMT2 knockdown cells that re-expressed the K95Q mutant compared with those that re-expressed WT SHMT2 (Fig. 5b), and this outcome was similar to what was observed in SHMT2 knockdown cells (Supplementary Fig. 4a). Taken together, these data indicate that acetylated SHMT2 cannot effectively utilize serine, which inhibits the rapid proliferation of colon cancer cells. Subsequently, the tumorigenicity of SHMT2 knockdown SW480 cells in which WT or K95Q mutant SHMT2 was re-expressed was analyzed in a xenograft model in vivo. As shown in Fig. 5c, the K95Q mutant-expressing xenograft tumors were significantly smaller in size than the WT SHMT2-expressing tumors, and this outcome is similar to what was observed in an SHMT2 knockdown xenograft model (Supplementary Fig. 4b). Since IDH2 also contributes to NADPH production and purine biosynthesis to facilitate cancer cell growth, we detected NADPH and ROS levels in colorectal cancer cells. We found that the loss of SHMT2 results in a similar phenotype as the re-expression of K95Q which significantly decreased NADPH and increased ROS in colorectal cancer cells (Fig. 5d, e and Supplementary Fig. 4c,

d). Additionally, we performed a formate rescue experiment in SHMT2 knockdown cells. We found that a formate supplement did not rescue the proliferation of SHMT2 knockdown cells (Supplementary Fig. 4e). These results demonstrate that SHMT2-K95-Ac impairs the ability of SHMT2 to support colon cancer cell growth and tumorigenicity through the attenuation of serine consumption and the reduction in the NADPH level. Moreover, we used the azoxymethane-dextran sodium sulfate (AOM-DSS)-induced colorectal tumorigenesis model in wild-type and Sirt3 KO mice. Sirt3 KO mice contained a significantly decreased number of colon tumors, and the expression level of SHMT2 was dramatically lower in Sirt3 KO cancer cells (Fig. 5f). Finally, we demonstrated that SHMT2 knockdown in SIRT3-overexpressing cells significantly attenuated the cell proliferation induced by SIRT3 overexpression (Supplementary Fig. 4f), and this outcome is similar to what was observed in a xenograft model (Supplementary Fig. 4g). The data demonstrate that SIRT3 promotes colorectal tumorigenesis through deacetylation and overexpression of SHMT2.

**K95-acetylation of SHMT2 is downregulated in CRC with high SIRT3 expression**. Our findings demonstrate that acetyl-mimetic substitution at lysine-95 impaired the ability of SHMT2 to support SW480 colon cancer cell proliferation and tumor growth. These results prompted us to examine K95 acetylation in SHMT2, and SHMT2, SIRT3, and TRIM21 protein expression levels in human colorectal cancer. We collected 35 pairs of primary human colorectal tumor samples to perform a direct immunoblotting analysis of the colorectal tumor samples (T) and their adjacent normal tissues (N). Of these 35 pairs of samples, 27 tumor samples exhibited a significant increase in the steady-state level of SHMT2 protein ($P < 0.001$), and the K95 acetylation level of SHMT2 was invariably reduced in all tumor samples with high SHMT2 expression (Fig. 6a, b). Interestingly, most tumor samples with high SHMT2 expression exhibited increased SIRT3 protein expression compared with adjacent normal tissues ($P < 0.01$). Additionally, we found inconsistent TRIM21 expression among the tumor samples (Fig. 6a, b and Supplementary Fig. 5). Strikingly, in our tissue pairs, SHMT2 protein levels were negatively correlated with K95 acetylation levels ($r = 0.3444$, $P = 0.0035$, Fig. 6c). Using an immunohistochemical analysis, we also revealed that levels of SHMT2 were positively correlated with levels of SIRT3 in 309 colorectal tumor samples. Quantification of the

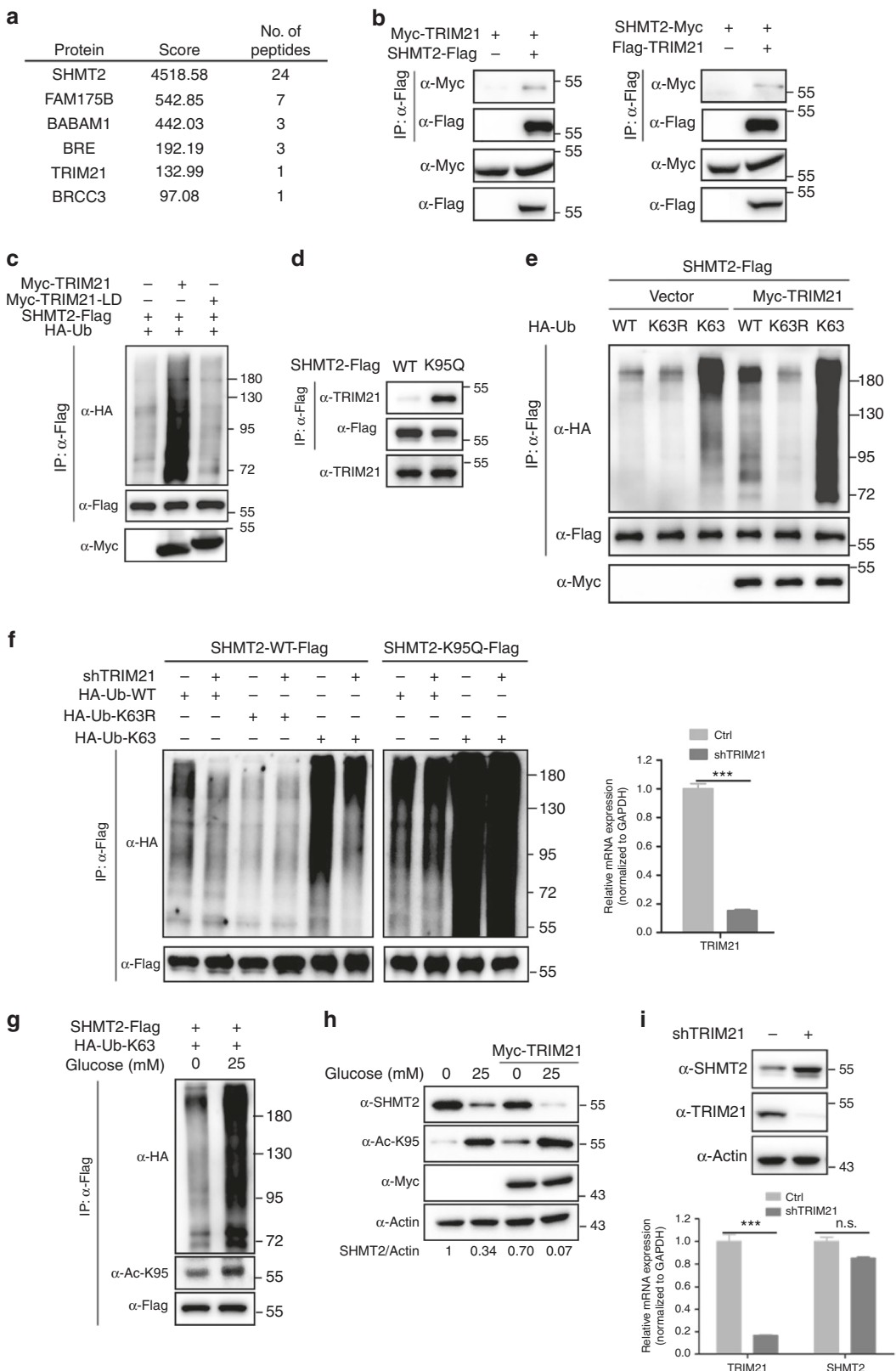

staining on a scale of 0–12 showed that these correlations were significant (r = 0.4662, P < 0.0001, Fig. 6d). Kaplan–Meier analysis indicated that the patients with higher SIRT3 expression (190 cases) tended to exhibit a poorer overall survival compared with patients with lower SIRT3 expression (119 cases)

(P = 0.0015) (Fig. 6e). Collectively, these data indicate that SHMT2 K95 acetylation is frequently downregulated in colorectal tumors and is associated with increased SIRT3 expression. Thus, SHMT2 K95 acetylation, SHMT2 protein and SIRT3 may be potential biomarkers for colorectal cancer.

**Fig. 4** TRIM21 is the E3 Ligase for SHMT2. **a** Identification of SHMT2-interacting proteins. SHMT2-interacting proteins involved in ubiquitination are shown. **b** TRIM21 interacts SHMT2. Interactions between ectopically expressed SHMT2 and TRIM21 in HEK293T cells were determined by western blot. **c** Wild-type TRIM21 not the ligase-dead (LD) mutant TRIM21 increases SHMT2 ubiquitylation. HA-Ub and SHMT2-Flag were co-transfected with Myc-TRIM21 WT or Myc-TRIM21-LD mutant in HEK293T cells. **d** K95Q mutant binds more TRIM21. Immunoprecipitation was performed from SHMT2-knockdown HeLa cells with either WT or K95Q mutant SHMT2-Flag stably rescued before being subjected to western blot with endogenous TRIM21 antibody. **e** TRIM21 increases K63-Ub chains of SHMT2. The indicated plasmids were transfected into HEK293T cells. **f** TRIM21 knockdown decreases K63-only-Ub not K63R-Ub chains of SHMT2. The indicated plasmids were transfected into TRIM21 knockdown or control HEK293T cells. The knockdown efficiency of TRIM21 was probed by Real-time PCR. **g** High glucose increases SHMT2 K63 poly-Ub chains. SHMT2-Flag and HA-Ub-K63-only were co-transfected into HCT116 cells for 36 h and then medium was replaced with containing 0 mM or 25 mM glucose medium for 6 h before harvest. **h** TRIM21 exacerbates SHMT2 degradation under high glucose. Vector or Myc-TRIM21 was transfected into HCT116 cells for 24 h and then medium was replaced with containing 0 mM or 25 mM glucose medium for 12 h before harvest. **i** TRIM21 knockdown increases endogenous SHMT2 protein. TRIM21 was stably knocked down in HCT116 cells by shRNA. The knockdown efficiency and SHMT2 protein level were determined by western blot. The mRNA level change of TRIM21 and SHMT2 was determined by Real-time PCR. For **f** and **i**, Mean values ± s.d. of relative mRNA expression of triplicate experiments are presented. ***$P < 0.001$

## Discussion

One-carbon metabolism has emerged as a key metabolic node in rapidly proliferating cancer cells[6]. The alteration of physiological processes in cancer cells by differential one-carbon pathway usage may highlight new opportunities for selective therapeutic intervention[37]. Many one-carbon metabolic enzymes, including SHMT2, which is responsible for intracellular serine and glycine interconversion, have been reported to be highly expressed in cancer cells and tumor samples derived from patients. The expression of mitochondrial SHMT2, but not cytosolic SHMT1, is upregulated in multiple cancer microarray datasets[9,38]. These phenomena are consistent with our findings in colorectal cancer.

In this study, we observed that the expression level of SHMT2 is significantly increased in patient tumor samples and is correlated with poorer overall postoperative survival (Supplementary Fig. 6a–c). We elucidated the mechanism of post-translational modifications in the regulation of the protein levels of SHMT2 in tumorigenesis. Lysine acetylation regulates SHMT2 through the inhibition of its enzymatic activity and the destabilization of SHMT2 via TRIM21-mediated K63-ubiquitin-macroautophagy (Fig. 6f). Low glucose increases SHMT2 protein levels by stimulating SIRT3-dependent deacetylation in colorectal cancer cells. Our structural and molecular data revealed that acetylation at K95 disrupts the functional homotetrameric structure of SHMT2 and decreases its affinity toward the substrate L-serine. We also determined that acetylated SHMT2 leads to the prevention of serine consumption, the upregulation of the serine/glycine ratio and the inhibition of NADPH production, which inhibits colorectal cancer cell growth and tumorigenesis. Our data suggest that acetylation negatively regulates SHMT2 function and prevents serine consumption during cell proliferation, which is consistent with the findings of others[6,39,40].

In this study, we have demonstrated that SHMT2-K95-Ac promotes its degradation via the K63-ubiquitin–lysosome pathway in a glucose-dependent manner. Specifically, after the use of the sirtuin inhibitor NAM or after SIRT3 knockout, SHMT2-K95-Ac is enriched in the cytoplasm, triggering SHMT2 degradation through lysosome-mediated macroautophagy. TRIM21 belongs to the tripartite motif (TRIM) family and is an interesting novel gene, whose protein product is a (RING) finger domain–containing E3 ligase. Several reported substrates of TRIM21 are involved in innate and adaptive immunity, such as IRF5 and SQSTM1/p62. Our study demonstrated that SHMT2 is a novel substrate of TRIM21 that regulates SHMT2 protein stability in a K95-acetylation-dependent manner. Interestingly, TRIM21 has recently been reported to downregulate Par-4 and act as a potential therapeutic target in colon and pancreatic cancers[41]. Although we

did not observe that TRIM21 contributes to the growth of colorectal cancer cells (Supplementary Fig. 4h), or the protein level of TRIM21 is not significantly different between colorectal tumor samples and normal tissues (Fig. 6a, b), the activation of TRIM21 to degrade SHMT2 levels may still be a promising therapeutic strategy for colorectal tumors.

All three mitochondrial Sirtuins, SIRT3, SIRT4 and SIRT5, have closely linked with tumorigenesis. Due to commercial antibodies issue, we only detected SIRT3 and SIRT5 in our clinical tissues. Consistently with recent report by Wang Y. et al[42], SIRT5 was overexpressed in CRC tissues compared with their matched normal mucosa in our larger CRC cohort (Supplementary Fig. 6d). However, SIRT3 is the major NAD$^+$-dependent deacetylase in mitochondria, which has been extensively studied in tumorigenesis. SIRT3 plays a tumor suppressor role in hepatocellular carcinoma and gastric cancer, but it acts as an oncogene in breast cancer, lung cancer and bladder cancer[43]. In our study, we determined that SIRT3 plays an oncogenic role in colorectal cancer via the deacetylation of SHMT2, which results in increased SHMT2 protein and elevated serine consumption in tumor cells. Moreover, our clinical analysis showed that high expression levels of SIRT3 in colorectal cancer patients were correlated with a shorter survival time, which is partially due to deacetylated SHMT2 and activated serine consumption in colorectal cancer cells. Similar to our observations in this study of SHMT2 regulation by SIRT3, SIRT3 also deacetylates cytoplasmic Skp2 to destabilize the protein and to suppress the oncogenic function of Skp2[44]. Our results, therefore, provide further rationale for the development of SIRT3/SHMT2 antagonists for colorectal cancer therapy.

Targeting one-carbon metabolic enzymes, including SHMT2, is a potentially attractive approach to the design of innovative therapeutic drugs for cancer treatment. Indeed, methotrexate, a major class of cancer chemotherapy agents, has demonstrated the ability to bind and inhibit human SHMT2 in vitro[45]. However, drug resistance urges scientists to develop more effective and specific therapeutic interventions for various cancers. Other clinical trials of selective inhibitors that target SHMT2, including leucovorin (N5-CHO-THF)[46] and 3-bromopyruvate (3BP)[47], have been performed. Notably, leucovorin is a critical component of standard chemotherapy regimens used for the treatment of colorectal cancer[48]. Leucovorin increases the intracellular concentration of 5,10-methylene tetrahydrofolate ($CH_2THF$) and inhibits thymidylate synthase (TS) activity through 5-fluorouracil[49]. Therefore, inhibition of SHMT2 activity by leucovorin could also account for the anti-cancer effects. However, more effective and specific SHMT2 inhibitors for therapeutic intervention in various cancers need to be developed. Our study shows that K95-acetylation of SHMT2

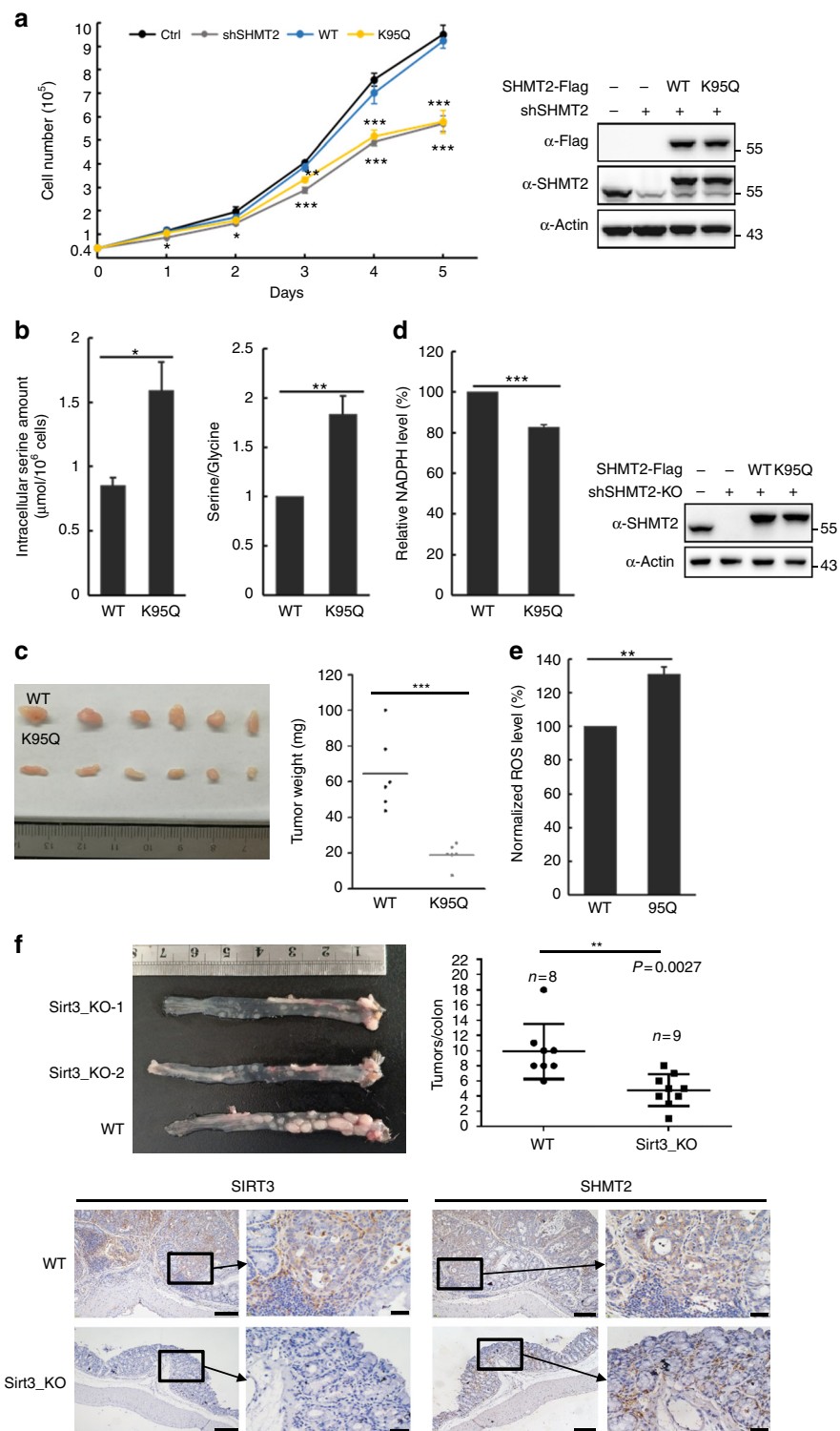

inhibits its function in converting serine to glycine, which results in impaired serine consumption in colorectal cancer cells and inhibits cell proliferation and tumor growth. This indicates an opportunity to develop a drug that targets SHMT2 K95 acetylation or activity for colorectal cancer treatment.

## Methods

**Antibodies and reagents**. The western blot primary antibodies to flag (Sigma SAB4301135, 1:5000), HA (Abcam 9110, 1:1,000), Myc (Cell Signaling Technology 2272, 1:1,000), Actin (Beyotime AA128, 1:5000), Acetylated-lysine (Cell Signaling Technology 9441, 1:1,000), SHMT2(Santa Cruz 25064, 1:1,000), SIRT3 (Cell Signaling Technology 2627, 1:1,000), TRIM21(Cell Signaling Technology 92043, 1:1,000), K63-Ub (Abcam 179434, 1:1,000), COXIV (Cell Signaling Technology 3E11, 1:2,000), ATG5 (Cell Signaling Technology D5F5U, 1:1,000), Tublin (Beyotime AT819-1, 1:5000) were commercially obtained. Antibody specific to SHMT2 K95Ac was prepared commercially from immunizing rabbits at Shanghai Youke Biotechnology (antigen peptide: CLNNK (Ac) YSEGY-NH2, 1:200). NAM (Sigma 72345), Cyclo-heximide (CHX)(Sigma 01810), MG132(Sigma M8699), NH4Cl (Sangon Biotech), 3-methyladenine (3-MA)(Selleck, CAS:5142-23-4), sodium formate (Sigma 71539) are commercially obtained.

**Fig. 5** SHMT2 acetylation inhibits cell proliferation and tumor growth. **a** SHMT2-K95Q is compromised to support cell proliferation. SW480 cells stably knockdown SHMT2 with re-express the shRNA-resistant wild-type or K95Q mutant were established. SHMT2 knockdown efficiency and re-expression were determined by western blot. SHMT2 WT or SHMT2 K95Q cells were seeded in 12-well plates. Cell numbers were counted every 24 h. **b** SHMT2 K95Q increases intracellular serine level and serine/glycine ratio. Intracellular serine and glycine amount were measured by GC-MS. **c** SHMT2 K95Q is defective in supporting tumor growth in vivo. Xenograft was performed using the SHMT2-knockdown SW480 cells with reexpressing wild-type or K95Q mutant SHMT2 as indicated. ***$P < 0.001$. **d** SHMT2 K95Q decreases cellular NADPH level. SHMT2 WT or K95Q-rescued HCT116-SHMT2-knockout cells were established and confirmed by western blot. Intracellular NADPH was determined by using NAD(P)H-Glo Detection System. **e** SHMT2 K95Q increases cellular ROS level. SHMT2 WT or K95Q-rescued HCT116-SHMT2-knockout cells were seeded in confocal dishes and ROS level was measured by adding 10 μM H2DCF-DA. Fluorescent strength per unit area was quantified using the ImageJ software, followed by statistical analysis. **f** Sirt3-knockout mice developed less tumors in the AOM-DSS CRC mouse models. Sirt3 WT and KO mice were subjected to AOM-DSS colitis-associated cancer model. The representative images of colon tumors are shown. Tumor number of each group were shown. $n = 8$ (WT), $n = 9$ (Sirt3_KO), $P = 0.0027$ by student's $t$-test. The expression of Sirt3 and Shmt2 in the colon tumors from WT and Sirt3_KO mice were detected by IHC. For **a**, **d**, and **e**, mean values ± s.d. of triplicate experiments are reported. *$P < 0.05$; **$P < 0.001$; ***$P < 0.001$

**Cell culture and transfection**. HEK293T, U2OS, SW480, SW620, HCT116, ZR-75-30, and HeLa cells were purchased from the American Type Culture Collection (ATCC). HeLa, U2OS, HEK293T, Sirt3$^{-/-}$ MEFs, SW620, SW480 and HCT116 cells were cultured in DMEM/ high glucose medium (HyClone) supplemented with 10% fetal bovine serum (BI), 100 units ml$^{-1}$ penicillin and 100 μg ml$^{-1}$ streptomycin (Sangon Biotech). ZR-75-30 cells were cultured in RPMI-1640 (GIBCO) with supplemented with 10% fetal bovine serum (BI), MEM NEAA (GIBCO), 100 units ml$^{-1}$ penicillin and 100 μg ml$^{-1}$ streptomycin (Sangon Biotech). Cells were maintained in an incubator at 37 °C, in a humidified atmosphere containing 5% CO$_2$. Cell transfection was performed using PEI (Polysciences) based on a 3:1 ratio of PEI (μg): total DNA (μg). Cell transfection for siRNA was carried out by Lipofectamine RNAiMAX according to the manufacturer's protocol. Synthetic siRNA oligo nucleotides were obtained commercially from Shanghai Genepharma Co, Ltd. siATG5: 5′-GGTTTGGACGAATTCCAACTTGTTT-3′; siATG7: 5′-GCTTTGGGATTTGACACATTT-3′.

**CRISPR-Cas9 knockout**. The sgRNA sequences targeting SHMT2 and SIRT3 were designed by CRISPR designer at http://crispr.mit.edu/. The guide sequence targeting Exon 2 of human SHMT2 and Exon 2 of human SIRT3 are shown.
SHMT2: 5′-GTTGCTGTGCTGAGCCCGAA -3′
SIRT3: 5′- GTGGGTGCTTCAAGTGTTGT-3′

**Immunoprecipitation and western blot**. Cells were lysed in 1% Nonidet P40 buffer containing 50 mM Tris-HCl (pH 7.5), 150 mM NaCl, and multiple protease inhibitors (PMSF 1 mM, Aprotinin 1 μg ml$^{-1}$, Leupeptin 1 μg ml$^{-1}$, Pepstatin 1 μg ml$^{-1}$, Na$_3$VO$_4$ 1 mM, NaF 1 mM, for acetylation experiments, TSA 2.5 mM and NAM 25 mM in addition, in final concentrations). For ubiquitin modification assay, cells were lysed in 1% SDS buffer (Tris-HCl pH 7.5, 0.5 mM EDTA, 1 mM DTT) after harvested and boiled for 10 min. For immunoprecipitation, the lysates were diluted 10-fold in Tris-HCl buffer. Cell lysates were incubated for 3 h at 4 °C with anti-flag M2 agarose (Sigma A2220) after removed debris by centrifuging at 4 °C, 13,000 rpm for 15 min. The beads which contained immunoprecipitates were washed three times with lysis buffer (but containing 0.4% Nonidet P40) and centrifuged at 2000 rpm for 2 min between each wash. Then beads were boiled and centrifuged at 4 °C before loading on sodium dodecyl sulfate polyacrylamide gel electrophoresis and transferred onto nitrocellulose membrane (GE Healthcare 10600002) for western blot analysis.

**Co-immunoprecipitation assays**. Total cells were lysed in BC100 buffer (20 mM Tris-HCl (pH 7.9), 100 mM NaCl, 0.2% NP-40 and 20% Glycerol) containing protease inhibitors (PMSF 1 mM, Aprotinin 1 μg ml$^{-1}$, Leupeptin 1 μg ml$^{-1}$, Pepstatin 1 μg ml$^{-1}$, Na$_3$VO$_4$ 1mM, NaF 1 mM) and 1 mM dithiothreitol. Whole lysates were incubated with 1 μg of mouse anti-SHMT2 (Santa Cruz 390641) or rabbit anti-SIRT3 (Cell Signaling Technology D22A3). Briefly, bounded proteins were eluted with 0.1 M glycine (pH 2.5) and then neutralized with 1 M Tris buffer to prevent disturbance of heavy chain (around 55 kDa). The elutions were analyzed by western blot.

**Cytoplasmic and mitochondrial fractionation**. Mitochondria isolation was performed as previously reported[50]. In briefly, cells were harvest with Isolation Buffer (225 mM mannitol, 75 mM sucrose, 0.1 mM EGTA and 30 mM Tris-HCl pH 7.4.), followed by homogenization. The homogenate was centrifuged at $600 \times g$ for 6 min to discard unbroken cells and nuclei (pellet). The supernatant was collected and centrifuged at $7000 \times g$ for 10 min. And then collect supernatant and resuspend pellet for another centrifugation at $7000 \times g$ for 10 min, respectively. To avoid cross contamination, repeated centrifugation was needed. Both the pellet (mitochondrial fraction) and supernatant (cytoplasmic fraction) were boiled separately in SDS sample buffer.

**SHMT2 activity assay**. In this assay, DL-β-phenylserine (Sigma 171603) was used as the substituted substrate and the production of benzaldehyde was followed by its strong UV absorbance at 279 nm. A typical assay mixture in a 96-well UV plate (Corning 3635) contained 50mM DL-β-phenylserine, 1 mM EDTA, 25 mM sodium sulfate, 50 μM PLP, 50 mM Hepes buffer (pH 8.0), and the enzyme. The rate of the appearance of the product, benzaldehyde, was measured at 25 °C at 279 nm by an EPOCH2 microplate reader (BioTek).

**Recombinant human SHMT2 expression and purification**. The cDNA encoding human mitochondrial SHMT2 (isoform 3; NCBI reference sequence NP_001159831.1) was cloned into a pET28a vector with an additional sumo tag after His tag in N-terminal and transformed into *E. coli* strain BL21 (DE3). Overnight cultures were subcultured 1:100 into 1 liter of LB containing kanamycin (50 μg ml$^{-1}$) and 100 μM PLP (Sigma P9255). Cultures were grown at 37 °C with shaking to an A600 of 0.6–0.8, induced with 0.4 mM IPTG. Cells were grown overnight at 18 °C, harvested by centrifugation, washed in PBS and resuspended in 30 ml of binding buffer (20 mM Tris-HCl pH 8.0, 500 mM NaCl and 30 mM imidazole,100μM PLP), PMSF (0.5 mM) was added just before lysis. Cells were lysed through high pressure disruption by a low temperature Ultra-high Pressure Continuous Flow Cell Disrupter (JN-3000 plus), and clarified cell lysate was obtained after centrifugation (17,000 rpm, 1 h, 4 °C). Samples were loaded onto a 1-ml HisTrap HP Ni column (GE Healthcare) by a peristaltic pump and gradient eluted by an ÄKTA FPLC system (GE Healthcare) with elution buffer (20 mM Tris-HCl pH 8.0, 500 mM NaCl and 500 mM imidazole, 100 μM PLP). Eluates were collected and performed ULP enzyme digestion to discard sumo tag protein meanwhile dialysis (dialysis buffer: 20 mM Tris-HCl pH 8.0, 500 mM NaCl, 100 μM PLP, 5% glycerol) for 3 h at 4 °C. Samples were ultrafiltration concentrated and loaded into a Superdex 200 16/600 FPLC column (GE Healthcare) and eluted using gel filtration buffer (20 mM Tris-HCl pH 7.4, 100 mM NaCl, 2 mM DTT). Recombinant SHMT2 K95R/Q mutant plasmid was generated by site-directed mutagenesis with a KOD-Plus-Mutagenesis kit (TOYOBO), expression and purification were conducted as described above.

**Crystallization and data collection**. The K95R mutant of SHMT2 was crystallized by hanging vapor-diffusion method by mixing 1 μl of 0.7 mg ml$^{-1}$ protein with 1 μl reservoir solution at 18 °C. The diffraction quality crystals of K95R were grown in a reservoir solution containing 0.2 M ammonium sulfate, 0.1 M Bis-Tris (pH 7.0), 20% (w/v) Polyethylene glycol 3350. All of the crystals were briefly soaked in a cryoprotectant solution consisting of 25% (v/v) glycerol dissolved in their corresponding mother liquors before being flash-cooled directly in a liquid-nitrogen stream at 100 K. The X-ray diffraction data were collected at the BL17U1 and BL18U1 beamlines of Shanghai Synchrotron Radiation Facility (Shanghai, China). Intensity data were integrated and scaled using HKL3000.

**Structure determination and refinement**. The structure of K95R was determined by molecular replacement using the SHMT2 structure (PDB code 4PVF) as the search model. Cycles of refinement and model building were carried out by using REFMAC and COOT programs until the crystallography R-factor and free R-factory values reached to satisfied range. The quality of the final model was evaluated with PROCHECK. All of the structures were displayed and analyzed using PyMOL program. The collected data and refinement statistics are summarized in Supplementary Table 1.

**Genetically encoding Nε-acetyl-lysine in recombinant proteins**. To generate a homogenously K95-acetylated SHMT2 construct, we used a three-plasmid (TEV-8, pCDFpylT-1, and pAcKRS) system as previously described. We cloned wild-type SHMT2 into pTEV-8 producing a C-terminal His6-tagged construct, and incorporated an amber codon at lysine 95 (AAG to TAG by site-directed mutagenesis). The amber construct was overexpressed in LB with spectinomycin

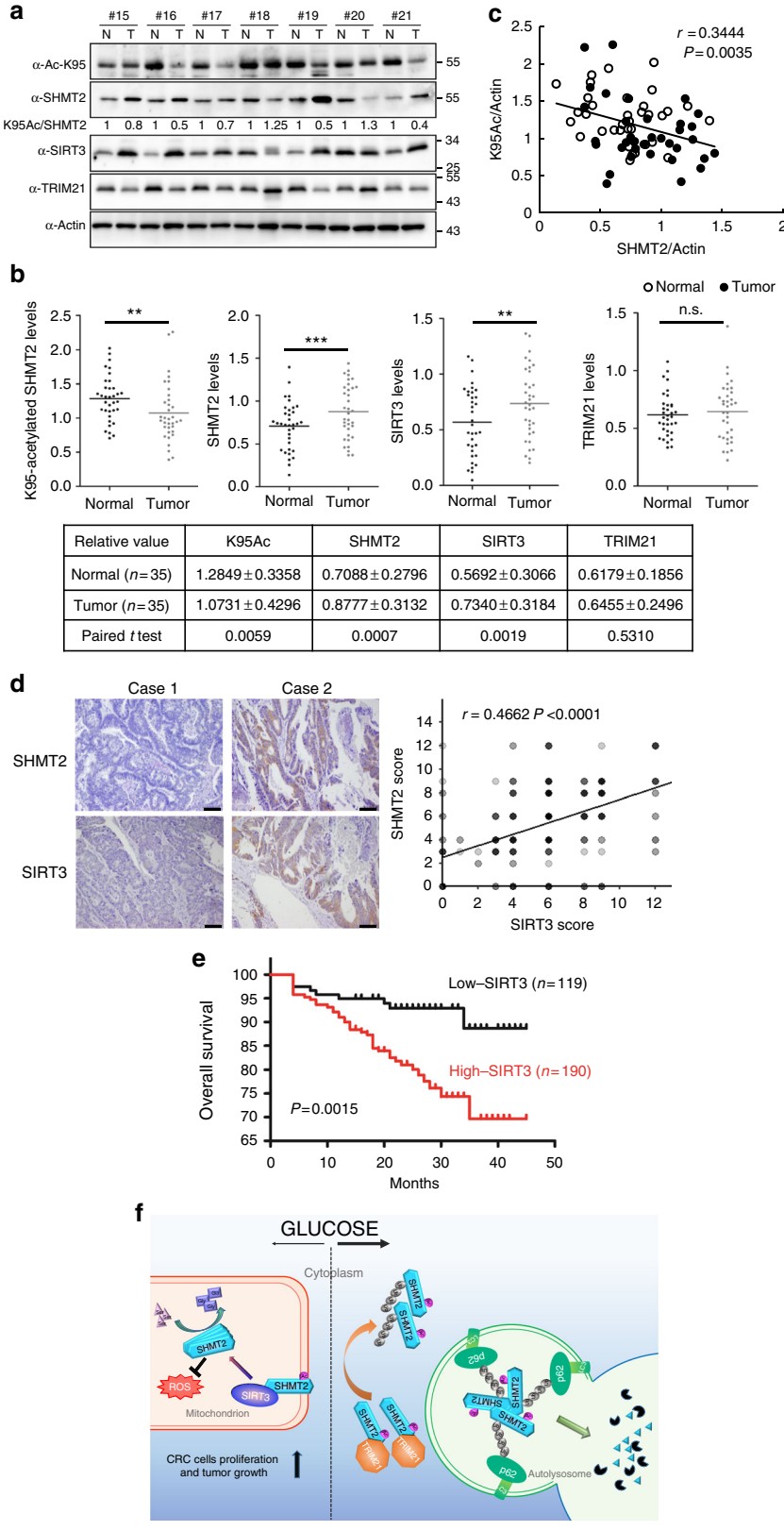

(50 mg ml⁻¹), kanamycin (50 mg ml), and ampicillin (150 mg ml⁻¹), in addition to 2 mM N-acetyl-lysine (Sigma A4021) and 20 mM nicotinamide to inhibit the activity of *E. coli* deacetylases at the time of induction. Cell culture, expression, and purification were referred to recombinant human SHMT2 methods as described above.

**In vitro SHMT2 deacetylation assay**. For the deacetylation experiments, recombinant K95-acetylated SHMT2 and recombinant human SIRT3 or inactive SIRT3-H248Y mutant were resuspended in 100 μl of deacetylation buffer (50 mM Tris-HCl pH 9.0, 4 mM MgCl$_2$, 50 mM NaCl, 0.5 mM DTT, 0.5 μM TSA) with or without 1 mM NAD$^+$. Reactions were incubated at 30 °C for 2.5 h and then placed

**Fig. 6** K95-acetylation of SHMT2 is downregulated in CRC. **a** In total, 35 pairs of tumor tissues (T) and adjacent normal tissues (N) were lysed. Protein levels of SHMT2, SHMT2-K95-Ac, SIRT3 and TRIM21 were determined by direct western blot. Relative protein levels were normalized by β-actin. Shown are seven pairs of samples. See Supplementary Fig. 6 for the other 28 pairs of samples. **b** Quantification of relative SHMT2, SHMT2-K95-Ac, SIRT3, and TRIM21 protein levels in the 35 pairs of samples tested. The intensities of indicated proteins were quantified using the ImageJ software, followed by statistical analysis. *$P < 0.05$; **$P < 0.001$; ***$P < 0.001$; n.s. not significant. **c** SHMT2 protein levels show negative correlation with SHMT2-K95-Ac. Correlation between SHMT2 protein levels and SHMT2-K95-Ac levels in the tested 35 pairs of samples. Statistical analyses were performed with $F$-test. **d** Positive correlation between SHMT2 and SIRT3 staining patterns in colorectal cancer. Three-hundred and nine colorectal cancer specimens were immunohistochemically stained with indicated antibodies. Representative photos of tumors were shown (left). Case 1, low SHMT2 expression in a low-SIRT3-expression specimen; Case 2, high SHMT2 expression in a high-SIRT3-expreesion specimen. Scale bars: 100 μm. Pearson correlation test was used to assess the statistical significance using R statistical program (right). Note that some of the dots on the graphs represent more than one specimen (i.e., some scores overlapped). **e** The patients with high SIRT3 expression ($n = 190$) have poorer overall survival compared with low-SIRT3 expression ($n = 119$). Significance was determined using Kaplan–Meier analyses. **f** Acetylation at K95 under high glucose inhibits SHMT2 enzyme activity and promotes its lysosomal degradation via macrophage. In CRC, low glucose promotes deacetylation of SHMT2 to stabilize SHMT2 and maintain its high activity, increasing cell proliferation and tumor growth

---

on ice for 15 min. For NAM treatment as a control, reactions were pretreated with 10 mM nicotinamide for 10 min at 30 °C.

**Colorectal cancer specimen collection**. All the human samples were collected in the Department of Colorectal Surgery, XinHua Hospital, from January 2008 to December 2016. Institutional review board approval and informed consent were obtained for all collections. Three-hundred and nine paired CRC and normal colon specimens were used to prepare tissue arrays for SHMT2 and SIRT3. Another 35 paired fresh samples were collected for the western blot analysis to analyze the SHMT2-K95-Ac, the expression level of SHMT2, SIRT3, and TRIM21.

**Immunohistochemistry**. The paraffin sections were deparaffinized, rehydrated, and treated according to standard protocol. After incubating with the indicated antibodies overnight, sections were washed three times with PBS and incubated with HRP-conjugated secondary antibody for 30 min at room temperature. Following three 5-min rinses in PBS, staining was developed with 3,3'-diaminobenzidine (DAB) solution for 10 min. The sections were then counterstained with 0.1% hematoxylin and sealed with coverslips.

Immunohistochemical staining was evaluated by assessing five representative fields per section at x200 magnifications with a light microscope (Carl Zeiss, Göttingen, Germany). Staining in tumor and normal tissues were scored according to the following standards: staining intensity was classified as 0 (lack of staining), 1 (mild staining), 2 (moderate staining) or 3 (strong staining); the percentage of staining was designated 1 ( < 25%), 2 (25–50%), 3 (51–75%), or 4 ( > 75%). For each section, the semi-quantitative score was calculated by multiplying these two values (which ranged from 0–12). Two histopathologists were blindly assigned to review the slides and score the staining.

**Quantitative real-time PCR**. Total RNAs were extracted from cells by using TRIzol reagent (Ambion) and then reverse-transcribed by using the qPCR RT Master Mix with gDNA Remover kit (TOYOBO). The products were then used as templates for real-time PCR using the SYBR Green Real-time PCR Master Mix (TOYOBO) on a CFX Connect Real-Time System (Bio-Rad). The primers used for different target genes were seen Supplementary Table 2.

**shRNA-knockdown, rescued, and overexpression cell pools construction**. TRIM21 and SHMT2 knockdown were generated using a lentivirus–mediated delivery system. The target sequences of shRNAs used in this study are as follows: shTRIM21: 5′-GAGTTGGCTGAGAAGTTGGAA-3′; shSHMT2: 5′-GTCTGACGTCAAGCGGATATC-3′. pMKO-shTRIM21, pMKO-shSHMT2 or pMKO-vector were co-transfected with pGAG and pVSVG into HEK293T cells. Retroviral supernatant was harvested 36 h after initial plasmid transfection and mixed with 8 μg ml⁻¹ polybrene to increase the infection efficiency. Cells pools were selected by 2 μg ml⁻¹ puromycin for 2 weeks. The knockdown efficiency was probed by Real-time PCR or western blot. In order to abrogate binding of SHMT2 shRNAs used in this study 6 silent mutations (highlighted in underline below) were introduced into the exogenous Flag-SHMT2 and Flag-SHMT2-K95Q cDNA sequence to produce a shRNA-resistant version. The GTCTGACGT-CAAGCGGATATC target sequence of shSHMT2 was converted to GTCA-GAT<u>G</u>T<u>G</u>AA<u>A</u>CGCAT<u>T</u>TC. shRNA-resistant flag-tagged human wild-type and K95Q mutant SHMT2 were cloned into the retroviral vector (pQCXIH), after retroviruses production, shSHMT2 stable cells in a state of suspension were infected with the prepared virus for 48 h and screened by 250 μg ml⁻¹ hygromycin for at least 2 weeks. TRIM21 and TRIM21-LD overexpression stable cell pools were generated using pBABE retrovirus system and SIRT3 overexpression using pQCXIH retrovirus system.

**GC-MS analysis**. Indicated stable SW480 cell lines were seeded in 60 cm plates (at appropriate seeding density to be ~80% confluent by the end of the assay). Cells were harvested with 80% pre-cold methanol and incubated in −80 °C for 24 h. The

supernatants were transferred to 1.5 ml glass bottles after centrifugation by 12,000 rpm, 5 min at 4 °C and dried by Concentrator Plus (Eppendorf). And then the dried extract was oximated by 20 mg ml⁻¹ methoxyamine hydrochloride (Sigma 226904) dissolved in pyridine at 70 °C for 60–90min, followed by derivatization at 30 °C for 30–60 min with 20 μl N-tert-Butyldimethylsilyl-N-methyltrifluoro-acetamide (Sigma 394882) in 80 μl pyridine. Filtrated samples were subjected to GC-MS analysis on GC-MSD System (Agilent Technologies). For each cell line, serine and glycine abundance were normalized to cell count.

**NADPH levels assay**. Intracellular NADPH were determined in HCT116 cells by using NAD(P)H-Glo Detection System (Promega G9061). Cells were trypsinized and lysed in extraction buffer followed by centrifugation at 12,000 rpm for 15 min. The clear supernatant extracts were processed according to the manufacturer's instructions. Luminescence was detected using EPOCH2 microplate reader (BioTek).

**Cellular ROS assay**. Intracellular ROS production was determined in HCT116 cells by using 2',7'-dichlorofluorescein diacetate (H2DCF-DA) (Sigma-Aldrich D6883). Cells were washed with PBS and incubated with 10 μM H2DCF-DA at 37 °C for 30 min to load the fluorescent dye according to the manufacturer's instructions. Then cells were washed with PBS twice and observed by fluorescence microscope (Ex.488 nm, Em.525 nm).

**Proliferation assay**. Cells were seeded in 12-well plates at $4 \times 10^4$ and allowed to adhere for 24 h. Triplicate wells were seeded for each experimental condition. Cells were trypsinized, resuspended in DMEM containing 10% FBS, and counted with a Cellometer (Nexcelom Bioscience) every day over a 5 days period.

**Xenograft studies**. SW480 stable cell lines with SHMT2 knockdown and re-expressed shRNA-resistant wild-type or K95Q mutant SHMT2 were prepared. In all, $2 \times 10^6$ cells in PBS were subcutaneously injected into each nude mice (male, 6–7 -week-old), purchased from SLAC. Shanghai. Three weeks later, all mice were sacrificed and tumors were dissected, photographed, and weighed. Mouse experiments were done in compliance with the Institutional Animal Care and Use Committee guidelines at Fudan University.

**AOM-DSS CRC mouse models**. Paired wide-type and Sirt3 KO male mice of the same parental generation were induced from 12 weeks old. Mice were weighed and intraperitoneally injected with AOM (10 mg kg⁻¹ body weight, dissolved in PBS) on day 1, followed by three cycles of DSS dissolved in drinking water as a stepwise increasing concentration of 1.25%–1.5%–1.75% from day 2. In every DSS cycle, mice drank DSS water for 7 days, and then stopped for 14 days before the next cycle. Mice were sacrificed on day 65. Tumors were calculated and photographed after longitudinal dissection of the colon. Then the whole colon was sectioned and stained.

**Statistical analysis**. Statistical analyses was performed using GraphPad Prism, Microsoft Excel 2013 or R statistical program as indicated in the figure legends. Unless specified, comparisons between groups were made by unpaired two-tailed Student's $t$-test. Differences were considered statistically significant if $P < 0.05$.

## Data availability
Structure factors have been deposited in the RCSB Protein Data Bank with the accession codes: 5X3V for SHMT2-K95R. The authors declare that all the other data supporting the findings of this study are available within this manuscript and its Supplementary Information or from the corresponding authors on request.

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

## Acknowledgements

We thank the members of Yu Lab for discussions throughout this study. We thank Dr. Shimin Zhao for helping with GC-MS experiments and facility center of State Key Laboratory of Genetic Engineering for assistance on MS-MS experiments. We thank Dr. Qunying Lei for supporting Sirt3 KO mice. We appreciate Dr. Ronggui Hu for his generous gift of HA-Ub Wild-type and various mutants plasmids. We thank Dr. Feng Qian for sharing with *Sirt3*$^{-/-}$ MEFs. This work was supported by the National Key Research and Development Program of China (2016YFA0500600), National Natural Science Foundation of China (31771545, 91749120, 81672517, 81874177), Shanghai Committee of Science and Technology Grants (16JC1404000) and Shanghai Pujiang Program (18PJ1407400). W.Y. was supported by China "Thousand Youth Talents" and the Program for Professor of Special Appointment (Eastern Scholar) at Shanghai Institutions of Higher Learning.

## Author contributions

Z.W., J.S., Y.T. and X.C. performed experiments and analyzed data; Z.W., J.Z., and J.L. worked the structural experiment and data analysis for K95 mutations; G.W., X.C., J.S.,

and L.C. analyzed clinical patients' data; C.Y.L. and W.Y. supervised the project, analyzed data; Z.W., C.Y.L., and W.Y. wrote the paper.

## Additional information

**Competing interests:** The authors declare no competing interests.

