## [Peer Review File · Nature Communications]

Reviewers' Comments:

Reviewer #1:

Remarks to the Author:

The Yu Laboratory – State Key Laboratory of Genetic Engineering and Collaborative Innovation Center for Genetics and Development, School of Life Sciences and Zhongshan Hospital, Fudan University, Shanghai, China - Deacetylation of Serine Hydroxymethyl-transferase 2 promotes Colorectal Carcinogenesis

In this manuscript the authors propose several seminal and new observations that SHMT2 is increased, beyond this protein's normal physiological levels, in several malignancies to support nucleotide synthesis and cell proliferation in colorectal cancer patients, which is correlated with poorer postoperative overall survival and tumor, node and metastasis (TNM) stage. In addition, and the most important discovery in this manuscript, is the authors very convincingly show data that SHMT2-K95-Ac, which is deacetylated by SIRT3, negatively regulates SHMT2 in colon cancer cells through a mechanism that disrupts the tetrameric structure of SHMT2. This was done using a newly developed anti- SHMT2-K95-Ac antibody and Ac and de-Ac mimic mutant SHMT2 genes. Lastly, convincing clinical data, from human colon cancer samples, implies a correlative relationship between increased SHMT2-K95-Ac levels, as well as decreased SIRT3 levels, and the risk of a poor overall clinical outcome in colon cancer. This part of the manuscript is highly recommended.

However, this manuscript also has several very significant issues that need to be addressed since they do not, a priori, support the authors' hypothesis and/or lack a mechanistic connection to acetylation. In this regard, first the author's present data showing that increased or decreased levels of SHMT2 predict for clinical outcome; however, the authors' hypothesis is that SHMT2 acetylation is the underlying mechanism. For example, the first data presented by the authors show that "higher SHMT2 expression (166 cases) were correlated with poorer overall survival compared with patients with lower SHMT2 expression", but the manuscript theme is about SHMT2 acetylation. I do understand that the authors are suggesting that decreased SHMT2 expression and acetylation should both decrease activity and that this correlates with a better patient outcome, but this is a weak scientific argument. Since the acetylation data is so convincing, it is unclear to this reviewer why the genomic data is included, and the authors should consider removing this data or moving it to the end of the manuscript since it does not really support the role of SHMT2-K95-Ac.

Second, the authors suggest, using somewhat simplistic molecular and cell biology experiments, that UHRF1 is a major downstream target of SHMT2 in its tumor promoting phenotype. The experiments to justify this idea are underdeveloped and unclear. I assume that the manuscript is about mitochondrial SHMT2 (mSHMT2), and it is unclear how the change in SHMT2 acetylation would alter the activity of a cytoplasmic protein and induce a tumor promoting phenotype. Immunohistochemical data analysis of CRC samples also showed that SHMT2 positively correlated with levels of UHRF1, but this is not rigorous data to make a mechanistic link. In addition, data should be shown correlating SHMT2-K95-Ac and UHRF1 levels.

Third, the authors are clearly suggesting that SHMT2-K95-Ac promotes its degradation via the K63-ubiquitin-lysosome pathway in the glucose dependent manner, and TRIM21 acts as E3 ubiquitin ligase for SHMT2. This raises two very significant concerns that must be addressed in some way: (1) TRIM21 is a mitochondrial protein and how does it interact with and direct degradation of mSHMT2? and (2) Is SHMT2 degraded in the cytoplasm or the mitochondria? If it is in the cytoplasm, then how does SIRT3 direct a process in the cytoplasm when SIRT3 deacetylation activity is restricted to the mitochondria? This is a very significant concern in this manuscript, and the authors need to either address this or remove it from the manuscript.

Lastly, the rationale for investigating SHMT2 as a protein regulated by acetylation seems

underdeveloped and is based on this one sentence: "Since SHMT2 protein levels are significantly increased in 3 primary CRC tissues compared with their paired adjacent normal tissues without changed mRNA levels (Figure 1C), which indicate that posttranslational modification could be involved in regulating SHMT2". This reviewer does not understand this sentence or these results. Based on these results the authors propose that protein degradation, due to a PTM, is a driving mechanism to explain this result. However, this is premature and there are multiple other possibilities and much more is required here in the way of results and more rigorous controls.

Finally, the manuscript contains other concerns that are addressed below.

Major Concerns:

1. The authors present the high expression SHMT2 data in the first section of the manuscript. It would be more appropriate to move this data to the end of the manuscript, so as to include all of the clinical data in one section, or remove this data.
2. In Figures one and two, the authors propose that SHMT2 deletion is mechanistically similar to SHMT2-K95-Ac. This may be correct, but it is not proven in the manuscript, and this must be addressed in some way. They need to either remove this or provide some data to prove that the tumor biology is very similar in cells lacking SHMT2 to cells expressing SHMT2-K95-Q, as well as cells overexpressing SIRT3.
3. In Figure 2 it is not clear how a mitochondrial protein effects the nuclear protein function. Second, same as above, a deletion of SHMT2 may be functionally different than acetylated SHMT2, and the authors need to present experiments to show that SHMT2-K95-Q exhibits a similar phenotype, using tumor growth experiments.
4. In page 5, Figure 2, these two paragraphs are confusing. The data in the first two paragraphs are mainly about deletion instead of acetylation, which are not very relevant to the hypothesis. In addition, the cell cycle experiments are open-ended, and no data is presented to mechanistically link a change in cell cycle profile to SHMT2-K95-Ac.
5. The authors use the phrase "Caloric restriction" in regards to tissue culture experiments, and this is not correct. CR is a diet, and tissue culture cells grow in media. In addition, the experiments are presented when cells are in low glucose, which should activate SIRT3, while high glucose should turn SIRT3 off. This is confusing since high nutrient conditions generally inhibit sirtuins. At the very least the authors need to provide control experiments, such as glucose uptake and IGF status, to explain this.
6. The authors could consider SHMT2-K95-Ac as the correct way to present the acetylation status of the protein, as well as in describing the antibody.
7. The quality of the western blots needs to be improved. In many panels of the figures, the intensity of the different panels is very different, and the background intensity of all the immunoblots should be very similar.

Overall, the authors suggested several mechanisms in this manuscript regarding K95 acetylation of SHMT2 and when the manuscript is focused on SHMT2-K95-Ac, the story is novel and potentially quite important. However, many other aspects of this manuscript are somewhat confusing and distracting that either does not directly address the overarching hypothesis, such as the role of UHFR1, or the role of TRIM21, a cytoplasmic protein, in regulating protein levels of a mitochondrial protein. In addition, many of the experiments lack controls or conclusions are made that are not, a priori, justified by the data presented.

Reviewer #2:

Remarks to the Author:

It is known that SHMT2 is upregulated in colorectal cancer, and Wei et al. Repeated this finding and reported that showed that SHMT2 promoted cell proliferation. The authors have also shown that SHMT2 was acetylated at K95 and identified SIRT3 as a deacetylase. Furthermore, the

authors described that acetylation of SHMT2 promotes its degradation by TRIM21. However, some of the conclusions in the paper were based on assumptions and more work needs to be done to have a clear view of how SIRT3/SHMT2/TRIM21 was involved in the colorectal carcinogenesis. Some of the data are not robust enough, e.g. data derived from the K95Ac antibody. The reviewer feels the manuscript does not meet the standard of Nature Communications.

Major points:

1. The authors showed that SHMT2 was acetylated at K95, different from the K280 and K464 that they have identified before. The authors have shown that acetylation of K280 and K464 is regulated by calorie restriction, which provided no cohesive connection to the claimed role of K95 acetylation in colorectal cancer. How is the acetylation of K95 regulated? What is the acetyltransferase for SHMT2?
2. The K95Ac antibody does not seem to be very specific, as there was still significant amount of acetylation detected in K95R mutant (Figure 3F). Therefore, the blotting using this antibody could be misleading as acetylation at other Lys residues could account for the changes in different samples or treatments.
3. In Figure 8, the correlation of acetylated SHMT2 with tumor is rather poor, with less than 20% changes between normal and tumor tissues. Instead, the upregulation of SIRT3 seems to correlate with tumor better. The authors should provide evidence that SIRT3 works through SHMT2 to promote cell proliferation and carcinogenesis.
4. The authors identified TRIM21 as a potential E3 ligase to degrade acetylated SHMT2 (Figure 6). A degradation experiment is needed for testing whether TRIM21 could accelerate the degradation of SHMT2. Also, what happens after overexpression of TRIM21 in the cancer cells? More evidence needs to be provided to show a clear relationship between TRIM21 and carcinogenesis.
5. From the experiments meant to show that K95Q and K95R both have dramatically decreased enzymatic activity (Figure 4), it seems that K95 might be directly involved in the catalytic activity of SHMT2. Also, lacking tetramer formation cannot be the only reason why K95R and K95Q were enzymatically dead as there was still significant amount of tetramer formed.

Minor points:

1. In Figure 1D, what is the knockdown efficiency of the two shRNAs against SHMT2?
2. In Figure 1 F, the authors need to show the absolute values of Serine and Glycine levels. Levels of NADPH and ROS also need to be determined in the cells.
3. In Figures 3I and 3J, K95Ac antibody should be used.
4. In Figures 5A and 5B, blotting with K95Ac is required. The band intensities of Figure 5B also need to be quantified.
5. The correlation between glucose starvation and SHMT2 acetylation at K95 is not well substantiated. Acetylation at K280 and K464 needs to be checked, as acetylation of these two sites increases under calorie-restricted conditions. Figure 5D shows that glucose somehow downregulates the protein levels of SIRT3. Why?
- 6 The quality in Figure 5E is not good. As the input of Myc-SIRT3 seems to be not even, it is hard to draw a conclusion from the pellet alone.
7. The claim that SHMT2 is degraded through macroautophagy is facade as they only relied on inhibitor chloroquine, which points to a possibility that the lysosomal pathway is involved. I.e. there is no direct evidence to demonstrate a direct change of autophagic activity.

Reviewer #3:

Remarks to the Author:

The mitochondrial serine catabolic enzyme SHMT2 catalyzes the rate-limiting step in one carbon unit metabolism and drives nucleotide synthesis and cancer cell proliferation. Although the transcriptional regulation of SHMT2 by Myc and HIF has been well established, the post-translational regulation of SHMT2 was largely unknown. This study discovered that SHMT2 enzyme activity and stability are regulated by acetylation. The authors identified that K95 is the main

acetylation site of SHMT2 and Sirt3 interacts with and deacetylate SHMT2 at K95 site. In addition, the paper demonstrates K95 acetylation inhibits SHMT2 enzyme activity by impairing the formation of tetrameric SHMT2 and promotes SHMT2 degradation through macroautophagy. Data are also presented showing that K95-acetylation of SHMT2 is downregulated in CRC with increased expression of SIRT3. The paper demonstrates SHMT2 is upregulated in colorectal cancer and promotes cancer tumor growth through upregulation of UHRF1. The discovery is novel and the data presented is solid and clear. Here are some comments that may help the authors to improve the manuscript.

Major

1. RNA-Seq analysis of SHMT2 knockdown cells identifies UHRF1 was downregulated by SHMT2 knockdown. However, the mechanism is not clear. In fact, this part is not quite relevant to the major finding of this paper. The author may consider taking Figure 2 out for investigation in the future.
2. It will be interesting to test whether adding formate rescue the proliferation of SHMT2 KD cells or SHMT2 K95Q cells. This should be able to elucidate whether the decreased proliferation is due to reduced purine biosynthesis.
3. NH₄Cl or 3-MA treatments may have off-target effect. How about knocking down Atg proteins to block autophagy? Or measure SHMT2 half life in ATG5 or ATG7 KO cells.

Minor

1. Besides SIRT3, SIRT4 and SIRT5 are also reported to express in mitochondrial. It's interesting to investigate if SIRT4 and SIRT5 are also overexpressed in CRC.
2. Since both K95R and K95Q disrupt tetrameric formation of SHMT2, K95R is not appropriate control to mimic deacetyl-modification.
3. Please quantify the relative SHMT2 decrease in Figure 5B
4. In page 7 line 3 (counting from the bottom), '...the K95-acetylated SHMT2 exhibited no protein activity (Figure 4D)' should be '...the K95-acetylated SHMT2 exhibited no enzyme activity'.

Reviewer #4:

Remarks to the Author:

The manuscript by Wei et al., "Deacetylation of Serine Hydroxymethyl-transferase 2 promotes Colorectal Carcinogenesis", elucidates the role of lysine acetylation in regulating SHMT2 activity and in colorectal cancer (CRC). Consistent with prior results, the authors find that SHMT2, the mitochondrial enzyme that interconverts glycine and serine to support the folate cycle and many downstream metabolic processes, is overexpressed in CRC clinical specimens. In cultured CRC cells, SHMT2 depletion reduces expression of cell cycle genes, particularly UHRF1. The authors identify a role for the mitochondrial sirtuin deacetylase SIRT3 in deacetylating SHMT2 at K95 to promote its activity. The authors find that K95 acetylation also affects SHMT2 stability, via polyubiquitination and macroautophagy-mediated degradation.

Overall, this is interesting and novel work that makes an important contribution. SHMT2 is a key player in one carbon metabolism, a set of key metabolic pathways that is the target of several approved active chemotherapies. This manuscript contributes important new insights into regulation of SHMT2 levels and activity. There is an enormous amount of data presented, most of

it quite convincing. Moreover, it may have therapeutic implications, in that targeting the SHMT2/SIRT3/TRIM21 axis could represent a useful means to inhibit one carbon metabolism in cancer, with potential utility in a wide variety of malignancies. With that said, I do have important suggestions for improvements to the manuscript:

1. Although the writing is clear enough, the manuscript is riddled with grammatical and spelling errors. I highly recommend that the authors employ the services of a professional editor.
2. The authors note that cell cycle was identified as a major target pathway of SHMT2, but only focus on UHRF1. Was there a rationale for believing that there is something specific about this particular gene? If not, I recommend that the authors examine expression of a few more cell cycle genes identified in their expression analysis in SHMT2 KD CRC cells.
3. Perhaps I missed this, but the authors should examine the multimerization status of the K95 mutants in CRC cells, and the multimerization of SHMT2 in SIRT3 KD and control cells, presumably by blue native electrophoresis or the like.
4. I am little confused by the authors' model that SHMT2 is degraded via macroautophagy, which we usually think of as a means by which cells degrade whole organelles such as mitochondria. Do the authors believe that this effect is specific for SHMT2? What happens to other mitochondrial proteins under their experimental conditions? Are they also degraded?
5. There is a doublet in the anti-Ack blot in Figure 3J. Which, or both, is SHMT2? Quantifying these results would be helpful.
6. The reduction in SHMT2 protein levels upon NAM treatment is fairly subtle (Fig. 5). The authors should repeat this study at least 3 times and show quantification with accompanying statistical analysis, as well as the primary data (Figs. 5A, B, D).
7. The authors indicate that they identified TRIM2 as a SHMT2 interacting protein via mass spec-based interactome analysis, but they do not show these data.
8. The authors should perform some basic epistasis analysis to test their model. For example, does TRIM21 KD rescue the growth defect associated with SHMT2 KD? Similarly, what about genetic interactions in cells between SHMT2 and SIRT3 KD and overexpression?

Reviewer#1 Expertise : Sirtuin, metabolism (Remarks to the Author):

The Yu Laboratory – State Key Laboratory of Genetic Engineering and Collaborative Innovation Center for Genetics and Development, School of Life Sciences and Zhongshan Hospital, Fudan University, Shanghai, China - Deacetylation of Serine Hydroxymethyl-transferase 2 promotes Colorectal Carcinogenesis

In this manuscript the authors propose several seminal and new observations that SHMT2 is increased, beyond this protein's normal physiological levels, in several malignancies to support nucleotide synthesis and cell proliferation in colorectal cancer patients, which is correlated with poorer postoperative overall survival and tumor, node and metastasis (TNM) stage. In addition, and the most important discovery in this manuscript, is the authors very convincingly show data that SHMT2-K95-Ac, which is deacetylated by SIRT3, negatively regulates SHMT2 in colon cancer cells through a mechanism that disrupts the tetrameric structure of SHMT2. This was done using a newly developed anti- SHMT2-K95-Ac antibody and Ac and de-Ac mimic mutant SHMT2 genes. Lastly, convincing clinical data, from human colon cancer samples, implies a correlative relationship between increased SHMT2-K95-Ac levels, as well as decreased SIRT3 levels, and the risk of a poor overall clinical outcome in colon cancer. This part of the manuscript is highly recommended.

We do appreciate the reviewer for the positive comments regarding the importance of this manuscript.

However, this manuscript also has several very significant issues that need to be addressed since they do not, a priori, support the authors' hypothesis and/or lack a mechanistic connection to acetylation.

We thank the reviewer for the constructive comments. Please find our point-to-point response below.

In this regard, first the author's present data showing that increased or decreased levels of SHMT2 predict for clinical outcome; however, the authors' hypothesis is that SHMT2 acetylation is the underlying mechanism. For example, the first data presented by the authors show that "higher SHMT2 expression (166 cases) were correlated with poorer overall survival compared with patients with lower SHMT2 expression", but the manuscript theme is about SHMT2 acetylation. I do understand that the authors are suggesting that decreased SHMT2 expression and acetylation should both decrease activity and that this correlates with a better patient outcome, but this is a weak scientific argument. Since the acetylation data is so convincing, it is unclear to this reviewer why the genomic data is included, and the authors should consider removing this data or moving it to the end of the manuscript since it does not really support the role of SHMT2-K95-Ac.

Response: Following reviewer's suggestion, we moved Figure 1 to Supplemental Figure 6, which is related to Figure 6 in this revised version.

Second, the authors suggest, using somewhat simplistic molecular and cell biology experiments, that UHRF1 is a major downstream target of SHMT2 in its tumor promoting phenotype. The experiments to justify this idea are underdeveloped and unclear. I assume that the manuscript is about mitochondrial SHMT2 (mSHMT2), and it is unclear how the change in SHMT2 acetylation would alter the activity of a cytoplasmic protein and induce a tumor promoting phenotype. Immunohistochemical data analysis of CRC samples also showed that SHMT2 positively correlated with levels of UHRF1, but this is not rigorous data to make a mechanistic link. In addition, data should be shown correlating SHMT2-K95-Ac and UHRF1 levels.

Response: According to the suggestions of reviewers 1 and 3, we agreed to remove Figure 2 from this manuscript.

Third, the authors are clearly suggesting that SHMT2-K95-Ac promotes its degradation via the K63-ubiquitin–lysosome pathway in the glucose dependent manner, and TRIM21 acts as E3 ubiquitin ligase for SHMT2. This raises two very significant concerns that must be addressed in some way: (1) TRIM21 is a cytoplasmic protein and how does it interact with and direct degradation of mSHMT2? and (2) Is SHMT2 degraded in the cytoplasm or the mitochondria? If it is in the cytoplasm, then how does SIRT3 direct a process in the cytoplasm when SIRT3 deacetylation activity is restricted to the mitochondria? This is a very significant concern in this manuscript, and the authors need to either address this or remove it from the manuscript.

Response: To address these two questions, we performed several experiments to test whether the SHMT2-K95-Ac is localized within the cytoplasm. We performed a subcellular fractionation assay, which involved the treatment of HCT116 cells with NAM or the knockout of SIRT3. As shown in the New Figure 1m, SHMT2-K95-Ac was strongly elevated in the cytoplasm of HCT116 cells after treatment with NAM and after SIRT3 knockout (Figure 1m), indicating that SHMT2-K95-Ac may be involved in SHMT2 degradation in the cytoplasm. Moreover, we demonstrated that the acetylation-mimic K95Q SHMT2 can bind more endogenous TRIM21 (Figure 4d). High glucose was observed to enhance the acetylation of SHMT2 and bind more Ub-K63 (new Figure 4g). Additionally, we specifically knocked down Atg5 or Atg7 and found that SHMT2 had accumulated (Figure 3h and Supplemental Figure 3d). We also observed that the SHMT2-K95Q mutant exhibited a strong binding affinity to p62, the macroautophagy receptor (Figure Supplemental Figure 3f). These data show that acetylation of SHMT2-K95 facilitates its own degradation through lysosome-mediated-macroautophagy. We also noted that SHMT2 is able to translocate to cytoplasm according to Dr. Lin Hening's paper (Cao J. et al, *BioRxiv* 2017). According to Dr. Wei Wenyi's paper (Inuzuka H. et al, *Cell* 2012, PMID 22770219), SIRT3 also deacetylates a cytoplasmic protein Skp2 in mitochondria and promote Skp2 nuclear translocation, which is similar to our observations in this study of SHMT2.

Lastly, the rationale for investigating SHMT2 as a protein regulated by acetylation seems underdeveloped and is based on this one sentence: "Since SHMT2 protein levels are significantly increased in all 8 primary CRC tissues compared with their paired adjacent normal tissues without changed mRNA levels (Figure 1C), which indicate that posttranslational modification could be involved in regulating SHMT2". This reviewer does not understand this sentence or these results. Based on these results the authors propose that protein degradation, due to a PTM, is a driving mechanism to explain this result. However, this is premature and there are multiple other possibilities and much more is required here in the way of results and more rigorous controls.

Response: According to the suggestion of the reviewer, we agreed to remove Figure 1c. We also reorganized the manuscript and focused on acetylation regulates SHMT2 stability in a macroautophagy way, which is positively correlated with SIRT3 expressing in CRC samples.

Major concerns:

1. The authors present the high expression SHMT2 data in the first section of the manuscript. It would be more appropriate to move this data to the end of the manuscript, so as to include all of the clinical data in one section, or remove this data.

Response: According to the suggestion of the reviewer, we moved Figure 1 to Supplemental Figure 6, which is related to Figure 6 in this revised version.

2. In Figures one and two, the authors propose that SHMT2 deletion is mechanistically similar to SHMT2-K95-Ac. This may be correct, but it is not proven in the manuscript, and this must be addressed in some way. They need to either remove this or provide some data to prove that the tumor biology is very similar in cells lacking SHMT2 to cells expressing SHMT2-K95-Q, as well as cells overexpressing SIRT3.

Response: According to the suggestions of the reviewers 1 and 3, we agreed to remove Figure 2 from this manuscript. We also moved Figure 1 to Supplemental Figure 6, which

is related to Figure 6.

To address whether SHMT2-K95Q has a phenotype that is similar to that of SHMT2 deletion, we generated SHMT2-knockdown cells and re-expressed SHMT2-K95Q in SHMT2-knockdown cells to perform cell proliferation assays (Figure 5a), serine/glycine quantification assay (Figure 5b and Supplemental Figure 4a) and a xenograft assay (Figure 5c and Supplemental Figure 4b) in nude mice. In all assays, SHMT2-K95Q exhibited the similar tumor biology as SHMT2-knockdown cells.

Figure 5a

New Figure 5b

New Supplemental Figure 4a

Figure 5c

New Supplemental Figure 4b

Moreover, according to the suggestion of the reviewer, we performed the cell proliferation assay and xenograft model in which SHMT2 was knocked down in cells that overexpressed SIRT3. We observed that SHMT2 knock down in SIRT3-overexpressing cells significantly attenuated cell proliferation (Supplemental Figure 4f and 4g).

New Supplemental Figure 4f

New Supplemental Figure 4g

Moreover, we used the DSS-induced colorectal tumorigenesis model in wildtype and SIRT3 KO mice. SIRT3 KO mice significantly decreased the numbers of colon tumors, and the expression level of SHMT2 was dramatically lower in SIRT3 KO tumors (New Figure 5f).

New Figure 5f

3. Second, same as above, a deletion of SHMT2 may be functionally different than acetylated SHMT2, and the authors need to present experiments to show that SHMT2-K95-Q exhibits a similar phenotype, using tumor growth experiments.

Response: To address whether SHMT2-K95-Ac has a phenotype that is similar to that of SHMT2 deletion, we generated SHMT2-knockdown cells and re-expressed SHMT2-K95Q in SHMT2-knockdown cells to perform cell proliferation assays (Figure 5a), serine/glycine quantification assay (Figure 5b and Supplemental Figure 4a) and a xenograft assay (Figure 5c and Supplemental Figure 4b) in nude mice. In all assays, SHMT2-K95Q exhibited the similar tumor biology as SHMT2-knockdown cells.

New Figure 5b

New Supplemental Figure 4a

Figure 5c

New Supplemental Figure 4b

4. In page 5, Figure 2, these two paragraphs are confusing. The data in the first two paragraphs are mainly about deletion instead of acetylation, which are not very relevant to the hypothesis. In addition, the cell cycle experiments are open-ended, and no data is presented to mechanistically link a change in cell cycle profile to SHMT2-K95-Ac.

Response: Again, according to the suggestions of the reviewers 1 and 3, we agreed to remove Figure 2 from this manuscript.

5. The authors use the phrase “Caloric restriction” in regards to tissue culture experiments, and this is not correct. CR is a diet, and tissue culture cells grow in media. In addition, the experiments are presented when cells are in low glucose, which should activate SIRT3, while high glucose should turn SIRT3 off. This is confusing since high nutrient conditions generally inhibit sirtuins. At the very least the authors need to provide control experiments, such as glucose uptake and IGF status, to explain this.

Response: According to the reviewer’s suggestion, we have removed the phrase “Caloric restriction” from the manuscript. We agree that high nutrient conditions generally inhibit sirtuins, agreeing with our presented data (new Figure 3d) that high glucose levels prevent SIRT3 expression, while low glucose levels increase SIRT3 expression.

New Figure 3d

6. The authors could consider SHMT2-K95-Ac as the correct way to present the acetylation status of the protein, as well as in describing the antibody.

Response: According to the reviewer's suggestion, we have replaced SHMT2 K95 acetylation into "SHMT2-K95-Ac" in the manuscript.

7. The quality of the western blots needs to be improved. In many panels of the figures, the intensity of the different panels is very different, and the background intensity of all the immunoblots should be very similar.

Response: Thank the reviewer for the suggestions to improve the quality of our work. According to the reviewer's suggestion, we have adjusted the background intensity of all the immunoblots. The quality of the western blots in this revised version has been largely improved.

Overall, the authors suggested several mechanisms in this manuscript regarding K95 acetylation of SHMT2 and when the manuscript is focused on SHMT2-K95-Ac, the story is novel and potentially quite important. However, many other aspects of this manuscript are somewhat confusing and distracting that either does not directly address the overarching hypothesis, such as the role of UHFR1, or the role of TRIM21, a cytoplasmic protein, in regulating protein levels of a mitochondrial protein. In addition, many of the experiments lack controls or conclusions are made that are not, a priori, justified by the data presented.

Response: We do appreciate the reviewer for the positive comments and constructive suggestions. With these extensive experimentation and clarification, the quality of the manuscript is significantly improved. I hope the reviewer will now be satisfied.

Reviewer#2 Expertise: metabolism, autophagy, CRC (Remarks to the Author):

It is known that SHMT2 is upregulated in colorectal cancer, and Wei et al. Repeated this finding and reported that showed that SHMT2 promoted cell proliferation. The authors have also shown that SHMT2 was acetylated at K95 and identified SIRT3 as a deacetylase. Furthermore, the authors described that acetylation of SHMT2 promotes its degradation by TRIM21. However, some of the conclusions in the paper were based on assumptions and more work needs to be done to have a clear view of how SIRT3/SHMT2/TRIM21 was involved in the colorectal carcinogenesis. Some of the data are not robust enough, e.g. data derived from the K95Ac antibody. The reviewer feels the manuscript does not meet the standard of Nature Communications.

Response: We do appreciate the reviewer for the constructive suggestions. With these extensive experimentation and clarification, the quality of the manuscript is significantly improved. Please find our point-to-point response below.

Major points:

1. The authors showed that SHMT2 was acetylated at K95, different from the K280 and K464 that they have identified before. The authors have shown that acetylation of K280 and K464 is regulated by calorie restriction, which provided no cohesive connection to the claimed role of K95 acetylation in colorectal cancer. How is the acetylation of K95 regulated? What is the acetyltransferase for SHMT2?

Response: We apologize for any confusion due to our inadequate writing. We have rewritten the sentences to make the manuscript more clear and logical. Another paper that we coauthored reported that K464 was hyperacetylated in SIRT3 KO mice (Hebert et al *Mol. Cell* 2012, PMID: 23201123) and that K280 was identified by an acetylation proteomics study (Elia et al *Mol. Cell* 2015, PMID: 26051181). However, these two sites were not primary acetylation sites in our study (Supplemental figure 1a). We then performed mass spectrometry and identified K95 as a new acetylation site (Figure 1b and 1d). Moreover, we revealed that K95 is the primary functional acetylated site that affects SHMT2 degradation under varying glucose levels (Figure 3a-d).

Supplemental Figure 1a

Figure 1b

Figure 1d

Our data revealed that the acetylation of SHMT2-K95 is a response to changes in glucose concentration (Figure 3c and 3d). Moreover, no acetyltransferase has been recognized in mitochondria, where the local pH is higher, and the high concentration of acetyl-CoA favors nonenzymatic acetylation reactions (Eric Verdin and Melanie Ott, *Nat Rev Mol Cell Biol.* 2015, PMID: 25549891).

2. The K95Ac antibody does not seem to be very specific, as there was still significant amount of acetylation detected in K95R mutant (Figure 3F). Therefore, the blotting using this antibody could be misleading as acetylation at other Lys residues could account for the changes in different samples or treatments.

Response: We thank the reviewer for this suggestion. Actually, we believe the band detected for K95R mutant (original Figure 3F) is the endogenous acetylated SHMT2-K95 pulled down by the overexpression of SHMT2-Flag since SHMT2 usually forms a tetramer. To address the specificity of the SHMT2-K95-Ac antibody, we performed another experiment to knock out endogenous SHMT2. We then overexpressed wildtype or K95Q mutant prior to antibody detection. We found that the specificity of the SHMT2-K95-Ac antibody was sufficient (New Figure 1e). Moreover, we performed a dot-blot analysis to confirm that the SHMT2-K95-Ac antibody only recognized acetylated SHMT2-K95 peptide (new Supplemental Figure 1c).

3. In Figure 8, the correlation of acetylated SHMT2 with tumor is rather poor, with less than 20% changes between normal and tumor tissues. Instead, the upregulation of SIRT3 seems to correlate with tumor better. The authors should provide evidence that SIRT3 works through SHMT2 to promote cell proliferation and carcinogenesis.

Response: We agree that the correlation of acetylated SHMT2 with tumor is not so obvious. This could be due to the acetylation of SHMT2 promote SHMT2 degradation, which result in the changes of acetylated SHMT2 between normal and tumor tissues is relatively small. However, in our 35 paired normal and CRC tissues, statistically significantly decreased acetylated SHMT2 was still observed in a total 35 tumor samples (Figure 6b).

Figure 6b

To further explore the SIRT3-SHMT2 axis in colorectal carcinogenesis, we detected the expression of SIRT3 and SHMT2 in the AOM-DSS CRC mouse model. The lack of SIRT3 and low expression of SHMT2 were observed in normal colon epithelial cells. The expression of both SIRT3 and SHMT2 was increased in cancer cells (new Figure 5f). SIRT3 KO mice developed much fewer tumors compared with the SIRT3 WT mice, and the SHMT2 expression level was also lower in SIRT3 KO cancer cells (new Figure 5f).

New Figure 5f

Moreover, we demonstrated that the knock down of SHMT2 in cells that overexpressed SIRT3 significantly rescued the cell proliferation (new Supplemental Figure 4f and 4g).

New Supplemental Figure 4f

New Supplemental Figure 4g

4. The authors identified TRIM21 as a potential E3 ligase to degrade acetylated SHMT2 (Figure 6). A degradation experiment is needed for testing whether TRIM21 could accelerate the degradation of SHMT2. Also, what happens after overexpression of TRIM21 in the cancer cells? More evidence needs to be provided to show a clear relationship between TRIM21 and carcinogenesis.

Response: To address this issue, we have tested whether TRIM21 could accelerate the degradation of SHMT2 in high glucose conditions. We found that the overexpression of TRIM21 increased the degradation of SHMT2 in high glucose conditions by binding more ubiquitin (new Figure 4g). Moreover, overexpression of TRIM21 can accelerate the degradation of SHMT2 in high glucose conditions, as presented in new Figure 4h.

To address the relationship between TRIM21 and colorectal carcinogenesis, we overexpressed wildtype and mutant TRIM21 in CRC cells. We found that neither wildtype nor mutant TRIM21 affected the proliferation of CRC cells (new Supplemental Figure 4h), agreeing with our clinical data presented in Figure 6a-b. These data have addressed this reviewer's question that TRIM21 might not affect colorectal carcinogenesis because of other targets besides SHMT2.

5. From the experiments meant to show that K95Q and K95R both have dramatically decreased enzymatic activity (Figure 4), it seems that K95 might be directly involved in the catalytic activity of SHMT2. Also, lacking tetramer formation cannot be the only reason why K95R and K95Q were enzymatically dead as there was still significant amount of tetramer formed.

Response: We agree with the reviewer's comment and have revised this portion of the manuscript. In mammals, SHMT2 is active only as a homotetramer (Giardina G. et al, FEBS 2015, PMID 25619277), and destroying tetramer formation can dramatically decrease its enzymatic activity. K95 is located nearby the substrate-binding cleft (residues

99–109). In our study, a steady-state kinetic analysis of SHMT2 showed that K95 mutants exhibited decreased affinity for the substrate L-serine (Figure 2c). As shown in the crystal structure data (Figure 2e), compared with K95, R95 has more hydrogen bonds and Q95 has fewer hydrogen bond interactions with their binding partner, indicating that both lose appropriate space for effective substrate insertion. Therefore, K95 is important for the enzymatic activity of SHMT2 due to maintenance of tetramer formation and affinity toward the substrate L-serine.

Minor points:

1. *In Figure 1D, what is the knockdown efficiency of the two shRNAs against SHMT2?.*

Response: According to the suggestions of the reviewers 1 and 3, we removed Figure 1D from this manuscript. Similar to this knockdown of SHMT2, we performed knockdown SHMT2 in SW480 cells in Figure 5a. The knockdown efficiency of this shRNA against SHMT2 in SW480 cells was significantly high (Figure 5a).

2. *In Figure 1 F, the authors need to show the absolute values of Serine and Glycine levels. Levels of NADPH and ROS also need to be determined in the cells.*

Response: We thank the reviewer for pointing this out and have added this information to the revised manuscript (new Figure 5b and supplemental Figure 4a). We also detected NADPH and ROS levels in colorectal cancer cells. We found that the loss of SHMT2 led to a phenotype similar to that of K95Q re-expression after the knockout of SHMT2, which significantly decreased NADPH and increased ROS in colorectal cancer cells (new Figure 5d, 5e and Supplemental Figure 4c-d).

3. In Figures 3I and 3J, K95Ac antibody should be used.

Response: We thank the reviewer for pointing this out and have detected acetylated SHMT2-K95 in these experiments (new Figure 1h and 1i).

4. In Figures 5A and 5B, blotting with K95Ac is required. The band intensities of Figure 5B also need to be quantified.

Response: We thank the reviewer for pointing this out and have detected acetylated SHMT2-K95 in these experiments (new Figure 3a and 3b).

New Figure 3a

New Figure 3b

5. The correlation between glucose starvation and SHMT2 acetylation at K95 is not well substantiated. Acetylation at K280 and K464 needs to be checked, as acetylation of these two sites increases under calorie-restricted conditions. Figure 5D shows that glucose somehow downregulates the protein levels of SIRT3. Why?

Response: Again, we apologize for any confusion due to our inadequate writing. We have rewritten the sentences to make the manuscript more clear and logical. Another paper that we coauthored reported that K464 was hyperacetylated in SIRT3 KO mice (Hebert et al *Mol. Cell* 2012, PMID: 23201123) and that K280 was identified by an acetylation proteomics study (Elia et al *Mol. Cell* 2015, PMID: 26051181). However, these two sites were not primary acetylation sites in our study (Supplemental Figure 1a). We then performed mass spectrometry and identified K95 as a new acetylation site (Figure 1b). Moreover, we revealed that K95 is the primary functional acetylated site (Figure 1d).

Supplemental Figure 1a

Figure 1b

Figure 1d

Previous several studies (Hirschey M. et al, *Nature* 2010, PMID: 20203611; Someya S. et al, *Cell* 2010, PMID: 21094524) reported that high nutrient conditions generally inhibit sirtuins, agreeing with our presented data (new Figure 3d) that high glucose levels prevent SIRT3 expression, while low glucose levels increase SIRT3 expression.

New Figure 3d

6. The quality in Figure 5E is not good. As the input of Myc-SIRT3 seems to be not even, it is hard to draw a conclusion from the pellet alone.

Response: We thank the reviewer for pointing this out and have repeated the experiment and improved the quality of Figure 5E as the reviewer suggested (new Figure 3e).

New Figure 3e

7. The claim that SHMT2 is degraded through macroautophagy is facade as they only relied on inhibitor chloroquine, which points to a possibility that the lysosomal pathway is involved. I.e. there is no direct evidence to demonstrate a direct change of autophagic activity.

Response: Actually, in our work, we used NH_4Cl , not chloroquine, to inhibit lysosomes and found that SHMT2 is enriched after treatment with this lysosome inhibitor (Figure 3f). Furthermore, we found that serum starvation did not accelerate SHMT2 degradation, which indicated that SHMT2 degradation did not occur through chaperone-mediated autophagy. To further clarify the degradation of SHMT2 through macroautophagy, we treated cells with macroautophagy inhibitor 3-MA. We found that SHMT2 accumulated under this treatment condition (Figure 3g). Moreover, we specifically knocked down Atg5 or Atg7 and observed SHMT2 had also accumulated (new Figure 3h and Supplemental Figure 3d). Additionally, we observed that the SHMT2-K95Q mutant has a strong binding affinity to p62, the macroautophagy receptor (new Figure Supplemental Figure 3f). These data have addressed this reviewer's question as to whether acetylation of SHMT2 at K95 facilitates its degradation through lysosome-mediated-macroautophagy.

New Figure 3h

New supplemental Figure 3d

New supplemental Figure 3f

Reviewer #3 Expertise: Serine metabolism (Remarks to the Author):

The mitochondrial serine catabolic enzyme SHMT2 catalyzes the rate-limiting step in one carbon unit metabolism and drives nucleotide synthesis and cancer cell proliferation. Although the transcriptional regulation of SHMT2 by Myc and HIF has been well established, the post-translational regulation of SHMT2 was largely unknown. This study discovered that SHMT2 enzyme activity and stability are regulated by acetylation. The authors identified that K95 is the main acetylation site of SHMT2 and Sirt3 interacts with and deacetylate SHMT2 at K95 site. In addition, the paper demonstrates K95 acetylation inhibits SHMT2 enzyme activity by impairing the formation of tetrameric SHMT2 and promotes SHMT2 degradation through macroautophagy. Data are also presented showing that K95-acetylation of SHMT2 is downregulated in CRC with increased expression of SIRT3. The paper demonstrates SHMT2 is upregulated in colorectal cancer and promotes cancer tumor growth through upregulation of UHRF1. The discovery is novel and the data presented is solid and clear. Here are some comments that may help the authors to improve the manuscript.

We thank the reviewer for the positive comments regarding the reliability of presented data and the importance of this manuscript. We thank the reviewer for the constructive comments. Please find our point-to-point response below.

Major:

1. RNA-Seq analysis of SHMT2 knockdown cells identifies UHRF1 was downregulated by SHMT2 knockdown. However, the mechanism is not clear. In fact, this part is not quite relevant to the major finding of this paper. The author may consider taking Figure 2 out for investigation in the future.

Response: According to the suggestions of reviewers 1 and 3, we agreed to remove Figure 2 from this manuscript.

2. It will be interesting to test whether adding formate rescue the proliferation of SHMT2 KD cells or SHMT2 K95Q cells. This should be able to elucidate whether the decreased proliferation is due to reduced purine biosynthesis

Response: To address this issue, we performed formate rescue experiment in the cells which SHMT2 was knocked down. We found that formate supplement did not rescue the cellular proliferation of SHMT2 knockdown (new Supplemental Figure 4e). These data indicate that SHMT2 knockdown decreases CRC proliferation not due to reducing purine biosynthesis.

3. *NH4Cl* or 3-MA treatments may have off-target effect. How about knocking down *Atg* proteins to block autophagy? Or measure *SHMT2* half life in *ATG5* or *ATG7* KO cells.
 Response: To address this issue, we performed the knockdown of *Atg5* or *Atg7* in CRC to check the accumulation of *SHMT2*. We found that *SHMT2* protein was increased in the *Atg5* or *Atg7* knockdown cells (new Figure 3h, new Supplemental Figure 3d).

Minor:

1. Besides *SIRT3*, *SIRT4* and *SIRT5* are also reported to express in mitochondrial. It's interesting to investigate if *SIRT4* and *SIRT5* are also overexpressed in CRC.

Response: The expressions of *SIRT4* and *SIRT5* in CRC has been previously reported. M Miyo M. et al. (*BJC* 2015, PMID: 26086877) found that *SIRT4* expression was decreased in the CRC, which functions as a tumor suppressor in CRC. Wang Y. et al. (*Nature Communications* 2018, PMID: 29416026) reported that *SIRT5* was overexpressed at both the mRNA and protein levels and was a poor prognostic marker in CRC. We tested several commercial *SIRT4* antibodies, but none were suitable for IHC assay. Thus, we only detected the protein expression of *SIRT5* in our CRC tissue array by IHC, as the reviewer suggested. Consistently, *SIRT5* was overexpressed in CRC tissues compared with their matched normal mucosa in our larger CRC cohort (new Supplemental Figure 6d). We have included the expression and functions of all three mitochondrial *SIRT*s in the Discussion section of the manuscript.

New supplemental Figure 6d

2. Since both *K95R* and *K95Q* disrupt tetrameric formation of *SHMT2*, *K95R* is not appropriate control to mimic deacetyl-modification.

Response: We agree with the reviewer's comment and have revised this portion of the manuscript. In mammals, *SHMT2* is active only as a homotetramer, and destroying tetramer formation can dramatically decrease its enzymatic activity. *K95* is located nearby the substrate-binding cleft (residues 99–109) (Giardina G. et al, *FEBS* 2015, PMID 25619277). As shown in the crystal structure data (Figure 2e), compared with *K95*, *R95* has more hydrogen bonds and *Q95* has fewer hydrogen bond interactions with their

binding partner, indicating that both lose appropriate space for effective substrate insertion. In our study, a steady-state kinetic analysis of SHMT2 showed that both K95 mutants exhibited decreased affinity for the substrate L-serine (Figure 2c). Therefore, in this SHMT2 study K95R is not appropriate control to mimic deacetyl-modification. We can only emphasize that the acetylation in K95 regulates the function of SHMT2 through disrupting the enzymatic activity of SHMT2 and promoting degradation of SHMT2.

3. Please quantify the relative SHMT2 decrease in Figure 5B

Response: We thank the reviewer for pointing this out and have revised in the Figure 3B as the reviewer suggested.

New Figure 3b

4. In page 7 line 3 (counting from the bottom), ‘...the K95-acetylated SHMT2 exhibited no protein activity (Figure 4D)’ should be ‘...the K95-acetylated SHMT2 exhibited no enzyme activity’.

Response: We thank the reviewer for pointing this out and have revised in the manuscript as the reviewer suggested.

Reviewer #4 Expertise: Lysine acetylation, Mass spectrometry (Remarks to the Author):

The manuscript by Wei et al., “Deacetylation of Serine Hydroxymethyl-transferase 2 promotes Colorectal Carcinogenesis”, elucidates the role of lysine acetylation in regulating SHMT2 activity and in colorectal cancer (CRC). Consistent with prior results, the authors find that SHMT2, the mitochondrial enzyme that interconverts glycine and serine to support the folate cycle and many downstream metabolic processes, is overexpressed in CRC clinical specimens. In cultured CRC cells, SHMT2 depletion reduces expression of cell cycle genes, particularly UHRF1. The authors identify a role for the mitochondrial sirtuin deacetylase SIRT3 in deacetylating SHMT2 at K95 to promote its activity. The authors find that K95 acetylation also affects SHMT2 stability, via polyubiquitination and macroautophagy-mediated degradation.

Overall, this is interesting and novel work that makes an important contribution. SHMT2 is a key player in one carbon metabolism, a set of key metabolic pathways that is the target of several approved active chemotherapies. This manuscript contributes important new insights into regulation of SHMT2 levels and activity. There is an enormous amount of data presented, most of it quite convincing. Moreover, it may have therapeutic implications, in that targeting the SHMT2/SIRT3/TRIM21 axis could represent a useful means to inhibit one carbon metabolism in cancer, with potential utility in a wide variety of malignancies. With that said, I do have important suggestions for improvements to the manuscript:

We do appreciate the reviewer for the positive comments regarding the convincement of our presented data and the importance of this manuscript. We thank the reviewer for the constructive comments. Please find our point-to-point response below.

1. Although the writing is clear enough, the manuscript is riddled with grammatical and spelling errors. I highly recommend that the authors employ the services of a professional editor.

Response: According to the suggestion of the review, we employed the service of the Nature Research Editing.

SPRINGER NATURE | Author Services

Nature Research Editing Service Certification

This is to certify that the manuscript titled Deacetylation of SHMT2 promotes Colorectal Carcinogenesis was edited for English language usage, grammar, spelling and punctuation by one or more native English-speaking editors at Nature Research Editing Service. The editors focused on correcting improper language and rephrasing awkward sentences, using their scientific training to point out passages that were confusing or vague. Every effort has been made to ensure that neither the research content nor the authors' intentions were altered in any way during the editing process.

Documents receiving this certification should be English-ready for publication; however, please note that the author has the ability to accept or reject our suggestions and changes. To verify the final edited version, please visit our verification page. If you have any questions or concerns over this edited document, please contact Nature Research Editing Service at support@as.springernature.com.

Manuscript title: Deacetylation of SHMT2 promotes Colorectal Carcinogenesis
Authors: WEI YU
Key: 059D-641E-B189-288F-4F9A

This certificate may be verified at secure.authorservices.springernature.com/certificate/verify.

2. The authors note that cell cycle was identified as a major target pathway of SHMT2, but only focus on UHRF1. Was there a rationale for believing that there is something specific about this particular gene? If not, I recommend that the authors examine expression of a few more cell cycle genes identified in their expression analysis in SHMT2 KD CRC cells.

Response: UHRF1 and CCND1 are the most differentially expressed genes in our RNA-

seq analysis of SHMT2 KD CRC cells. SHMT2 knockdown induced G1/S arrest in CRC cells and UHRF1 is the critical checkpoint regulator in G1/S transition, thus we previously focused on the UHRF1 as the primary downstream effector of SHMT2 in promoting cell cycle in CRC cells. According to the suggestions of reviewers 1 and 3, we removed Figure 2 from this manuscript. The mechanism of SHMT2 regulating UHRF1 expression and the functional relationship of UHRF1 mediating the function of SHMT2 will be extensively explored in future.

3. Perhaps I missed this, but the authors should examine the multimerization status of the K95 mutants in CRC cells, and the multimerization of SHMT2 in SIRT3 KD and control cells, presumably by blue native electrophoresis or the like.

Response: According to the suggestions of this reviewer, to address this issue, we performed glutaraldehyde crosslink experiments to examine the multimerization status of SHMT2 in HCT116 as the reviewer suggested. The tetramer status of K95Q mutant was disrupted in HCT116 cells comparing with wildtype SHMT2 (new Figure 2h). Moreover, the tetramer status of acetylated SHMT2 was also significantly decreased in SIRT3 KO HCT116 cells (new Figure 2h). These data further demonstrate that acetylated SHMT2-K95 destroys SHMT2 function by disrupting the tetramer status.

New Figure 2h

4. I am little confused by the authors' model that SHMT2 is degraded via macroautophagy, which we usually think of as a means by which cells degrade whole organelles such as mitochondria. Do the authors believe that this effect is specific for SHMT2? What happens to other mitochondrial proteins under their experimental conditions? Are they also degraded?

Response: Autophagy is primarily responsible for the degradation of long-lived proteins and entire organelles and maintains therefore intracellular homeostasis, and it also contributes to starvation and stress responses. Macroautophagy degrades larger structures such as organelles and protein aggregates. In this manuscript, we have demonstrated that acetylated SHMT2 translocates to the cytoplasm and is degraded in the lysosome. Moreover, we have identified that K95-acetylation disrupts the SHMT2 tetramer and forms aggregates, which are degraded by macroautophagy. To address this issue, we also checked the stability of Methylenetetrahydrofolate Dehydrogenase 2 (MTHFD2), another mitochondrial protein, in ATG5 or ATG7 knockdown cells. No significant alteration in MTHFD2 protein was observed in ATG5 or ATG7 knockdown cells, indicating that the

macroautophagy degradation of SHMT2 is not the degradation of whole mitochondria in this manuscript (new Supplemental Figure 3e).

5. There is a double in the anti-Ack blot in Figure 3J. Which, or both, is SHMT2? Quantifying these results would be helpful.

Response: We thank the reviewer for pointing this out and have revised in the manuscript as the reviewer suggested. The down band detected by mouse anti-Ack antibody (Cell Signaling Technology 9681) in original Figure 3J is heavy chain of antibody due to the secondary mouse antibody. We have repeated this experiment using rabbit anti-Ack antibody (Cell Signaling Technology 9441) and secondary rabbit antibody (new Figure 1i).

New Supplemental Figure 1i

6. The reduction in SHMT2 protein levels upon NAM treatment is fairly subtle (Fig. 5). The authors should repeat this study at least 3 times and show quantification with accompanying statistical analysis, as well as the primary data (Figs. 5A, B, D).

Response: We thank the reviewer for pointing this out and have revised in the manuscript as the reviewer suggested. We have repeated these experiments more than 3 times in deed and also we have quantified the changes in new Figure 3a, 3b and 3d.

New Figure 3b

New Figure 3d

7. The authors indicate that they identified TRIM2 as a SHMT2 interacting protein via mass spec-based interactome analysis, but they do not show these data.

Response: We thank the reviewer for pointing this out and have revised the mass spectrometry data in the manuscript as the reviewer suggested. Interestingly, our mass spec-based interactome also identifies the components of BRISC complex, which is de-K63Ub complex (new Figure 4a).

New Figure 4a

Protein	Score	No. of peptides
SHMT2	4518.58	24
FAM175B	542.85	7
BABAM1	442.03	3
BRE	192.19	3
TRIM21	132.99	1
BRCC3	97.08	1

8. The authors should perform some basic epistasis analysis to test their model. For example, does TRIM21 KD rescue the growth defect associated with SHMT2 KD? Similarly, what about genetic interactions in cells between SHMT2 and SIRT3 KD and overexpression?

Response: To address this issue, we have performed CRC proliferation assays as the reviewer suggested. Overexpressing SIRT3 significantly increased cell proliferation and then knockdown SHMT2 significantly counteracted this effect (new Supplemental Figure 4f and 4g). We found that neither wildtype nor mutant TRIM21 affected CRC cells proliferation (new Supplemental Figure 4h), which is consistent with our clinical data in Figure 6a and 6b. These data addressed this reviewer's question that TRIM21 might not

affect colorectal carcinogenesis due to other targets besides SHMT2. Moreover, to further explore the SIRT3-SHMT2 axis in colorectal carcinogenesis, we detected the expression of SIRT3 and SHMT2 in the AOM-DSS CRC mouse model. The lack of SIRT3 and low expression of SHMT2 were observed in normal colon epithelial cells. The expression of both SIRT3 and SHMT2 was increased in cancer cells (new Figure 5f). SIRT3 KO mice developed much fewer tumors compared with the SIRT3 WT mice, and the SHMT2 expression level was also lower in SIRT3 KO cancer cells (new Figure 5f).

New Supplemental Figure 4f

New Supplemental Figure 4g

New Supplemental Figure 4h

New Figure 5f

Reviewers' Comments:

Reviewer #1:

Remarks to the Author:

The reviewers have done a very fine job of addressing this reviewers concerns and added new data that i think significantly improves the manuscript. The manuscript is well-written and justified review, and with the new data and changes to the presentation the major claims are stronger. I think the data is novel and will be of significant interest to others in the community and the wider field? The statistical analysis is appropriate and the data is clearly more rigorous than the first presentation. Overall, i fine this manuscript acceptable.

Reviewer #2:

Remarks to the Author:

My concerns/criticisms have been adequately addressed.

Reviewer #3:

None

Reviewer #4:

Remarks to the Author:

The changes made to the manuscript by Wei and colleagues have greatly strengthened it. This work makes a substantial novel contribution to the study of SHMT2 and cancer, and will be of broad interest to the readership of Nature Communications. I do have two remaining suggestions for relatively small experiments that I think would add substantially to this work:

1. I couldn't find anywhere that the authors had measured SHMT2 mRNA expression in cells with or without SIRT3, or +/- NAM treatment. It is important to rule out the possibility that reduced SHMT2 levels occur at least in part transcriptionally.
2. The authors' contention that ac-SHMT2 is present in the cytosol would be greatly strengthened by immunofluorescence analysis of NAM-treated or SIRT3-depleted cells, with both the ac-SHMT2 and total SHMT2 antibodies.

We would like to thank the editors and reviewers for the consideration of our manuscript for publication in *Nature Communications*. I have enclosed the specific point-by-point responses to reviewer 4's comments.

Reviewer #1 (Remarks to the Author):

The reviewers have done a very fine job of addressing this reviewer concerns and added new data that i think significantly improves the manuscript. The manuscript is well-written and justified review, and with the new data and changes to the presentation the major claims are stronger. I think the data is novel and will be of significant interest to others in the community and the wider field? The statistical analysis is appropriate and the data is clearly more rigorous than the first presentation. Overall, i fine this manuscript acceptable.

Reviewer #2 (Remarks to the Author):

My concerns/criticisms have been adequately addressed.

Reviewer#3

Editorial note: Reviewer#3 expresses their satisfaction with the revised manuscript in the confidential comments to the editor.

Reviewer #4 (Remarks to the Author):

The changes made to the manuscript by Wei and colleagues have greatly strengthened it. This work makes a substantial novel contribution to the study of SHMT2 and cancer, and will be of broad interest to the readership of Nature Communications.

We do appreciate the reviewer for the positive comments regarding the importance of this manuscript.

I do have two remaining suggestions for relatively small experiments that I think would add substantially to this work:

- 1. I couldn't find anywhere that the authors had measured SHMT2 mRNA expression in cells with or without SIRT3, or +/- NAM treatment. It is important to rule out the possibility that reduced SHMT2 levels occur at least in part transcriptionally.*

Response: According to the suggestion of the reviewer, we performed the realtime-PCR assay of SHMT2 in cells which SIRT3 was knocked down or in cells with NAM treatment. We observed that mRNA expression of SHMT2 was mildly increased in SIRT3-knocking down cells (new Supplemental Fig.1e), which is opposite to the downregulation of protein levels of SHMT2. Moreover, the mRNA expression of SHMT2 was not changed in cells with NAM treatment (new Supplemental Fig.3a). These data have ruled out the transcriptional regulation of SHMT2 levels in cells with SIRT3 deletion or NAM treatment.

new Supplemental Fig.1e

new Supplemental Fig.3a

2. *The authors' contention that ac-SHMT2 is present in the cytosol would be greatly strengthened by immunofluorescence analysis of NAM-treated or SIRT3-depleted cells, with both the ac-SHMT2 and total SHMT2 antibodies.*

Response: We agree with the reviewer's suggestion. Before, we have tried very hard to test our generated SHMT2-K95-Ac antibody in immunofluorescence, immunocytochemistry and immunohistochemistry assays. Unfortunately, this SHMT2-K95-Ac antibody was not suitable for these assays. Again as shown in the Figure 1m, our SHMT2-K95-Ac was strongly elevated in the cytoplasm of HCT116 cells after treatment with NAM and after SIRT3 knockout (Figure 1m), indicating that SHMT2-K95-Ac may be involved in SHMT2 degradation in the cytoplasm.

Figure 1m